# A Multiple-Step, Randomly Delayed, Robust Cubature Kalman Filter for Spacecraft-Relative Navigation

Rongjun Mu [1], Yanfeng Chu [1,*], Hao Zhang [2] and Hao Liang [2]

1   School of Astronautics, Harbin Institute of Technology, Harbin 150001, China
2   Beijing Institute of Aerospace Systems Engineering, Beijing 100076, China
*   Correspondence: chuyanfeng_hit@163.com

**Abstract:** This study is focused on addressing the problem of delayed measurements and contaminated Gaussian distributions in navigation systems, which both have a tremendous deleterious effect on the performance of the traditional Kalman filtering. We propose a non-linear, multiple-step, randomly delayed, robust filter, referred to as the multiple-step, randomly delayed, dynamic-covariance-scaling cubature Kalman filter (MRD-DCSCKF). First, Bernoulli random variables are adopted to describe the measurement system in the presence of multiple-step random delays. Then, the MRD-DCSCKF uses the framework of the multiple-step randomly delayed filter, based on a state-augmentation approach, to address the problem of delayed measurements. Meanwhile, it depends on a dynamic-covariance-scaling (DCS) robust kernel to reject the outliers in the measurements. Consequently, the proposed filter can simultaneously address the problem of delayed measurements and inherit the virtue of robustness of the DCS kernel function. The MRD-DCSCKF has been applied to vision-based spacecraft-relative navigation simulations, where quaternions are adopted to represent spacecraft's attitude kinematics, and the attitude update is completed with quaternions and generalized Rodrigues parameters. Monte Carlo simulations have illustrated that MRD-DCSCKF is superior to other well-known algorithms by providing high-accuracy position and attitude estimations in an environment with different delay probabilities and/or different outlier-contamination probabilities. Therefore, the proposed filter is robust to delayed measurements and can suppress outliers.

**Keywords:** spacecraft relative navigation; cubature Kalman filter; randomly delayed measurements; robust; dynamic covariance scaling

## 1. Introduction

Formation flights, on-orbit servicing, asteroid exploration, rendezvous and docking, active debris removal, and other space-based missions commonly require high-precision position and attitude estimations [1]. Molli et al. has proposed a satellite navigation system that combines inter-satellite links and a batch-sequential filter, and it reconstructs high-quality orbits for the constellation nodes and provides high-level autonomous navigation for Mars exploration [2]. Anna et al. has designed the space exploration rover with a positioning and mapping system which is equipped with light detection and ranging instruments, and it could achieve safe navigation in rough terrain and avoid obstacles [3]. Andolfo et al. has processed stereo images captured during the rover's travel with 3D visual odometry, where 2D image keypoints are identified to estimate the relative position and attitude between every step. This method has the advantage of mitigating trajectory inconsistencies from dead-reckoning techniques [4]. Junkins et al. has developed a vision-based spacecraft-relative navigation system using a position sensing diode (PSD) [5]. The vision-based navigation (VISNAV) system has the virtues of light weight, small size, and low power consumption [6]. It is usually adopted to determine the relative positions and attitudes of spacecraft that are within several hundred meters from each other. Optical sensors and beacons are essential components of the VISNAV system, and they enable

optional vision or adaptive vision. An optical sensor consists of a wide-angle lens and a position-sensing diode. A PSD possesses the merits of measuring the intensity and position of a light-point simultaneously. In vision-based spacecraft-relative navigation systems, light-emitting diodes (LEDs) are mounted as optical beacons on the chief spacecraft, and the deputy spacecraft is equipped with PSD-based optical sensors. The energy emitted from the optical beacons is then focused on the PSD through a wide-angle lens, so the optical signal is transformed into an electrical signal through electronic processing to acquire the image information of the target. In the more sophisticated VISNAV systems, the PSD can be programmed to only recognize a specified light source [7]. In VISNAV systems, the relative motion equations are built in the local–vertical–local–horizontal (LVLH) coordinate frame to obtain the motion state of targets. Generally, the beacon position is defined and known in the chief spacecraft body frame, and the position vector from the deputy spacecraft with respect to chief spacecraft is defined within the LVLH coordinate frame. Typically, the research in this field has assumed that the chief body is consistent within the LVLH coordinate system as well. However, this assumption is not rigorous and is just a special case. To eliminate the above assumptions, an efficient method is to estimate the attitudes of the two spacecrafts with respect to the LVLH coordinate frame, and the relative attitudes could then be obtained as well.

The spacecraft's orientation relative to the reference coordinate frame can be determined through the estimation of its attitude [8]. Euler angles, rotation vectors, direction cosine matrices, and quaternions are usually used to describe spacecraft attitude. Specifically, quaternions are useful for describing spacecraft attitude kinematics due to their non-singularity and the bi-linear kinematics equation [9]. To ensure that quaternions meet the normalization constraint, we adopted quaternions for propagation and updating and introduced the generalized Rodrigues parameter (GRP) [10] to denote the local attitude error. Jitter, vibration, and multiple disruptions can degrade the measurements' quality [11]. Vision-based spacecraft-relative navigation systems are subjected to randomly delayed measurements, and measurement noise is always disturbed by outliers. These factors motivated us to develop a filter that could simultaneously handle randomly delayed measurements and outliers.

The Kalman filter (KF) is the most commonly used estimation technique for the assumption of linear systems and Gaussian noise [12]. However, many problems do not satisfy the linearity assumption, and non-linear systems are more common in practical engineering. Vision-based spacecraft-relative navigation uses the optical camera to obtain measurements, which are modeled with non-linear equations. Among various non-linear filters, the extended Kalman filter (EKF), as a broadly available method, linearizes non-linear systems using the first-order Taylor series expansions. Unfortunately, this can result in significant truncation errors, and the procedure of deriving the Jacobi matrix is tedious [13]. A family of sigma-point filters and particle filters can avoid the loss of high-order terms, and therefore, they exhibit better performances than the EKF. The sigma-point filter approximates the probability density distributions (PDFs) of states through a group of defined sigma points and propagates the mean and covariance of random variables through non-linear processes. Julier has proposed the unscented Kalman filter (UKF) based on an intuitive statistical information transformation [14]. Arasaratnam adopted the spherical-radial cubature rule to calculate the means and covariances of state variables after non-linear propagation and proposed the cubature Kalman filter (CKF) [15]. Jia proposed the high-degree cubature Kalman filter (HCKF) based on the Genz's code [16] and the generalized Gauss–Laguerre quadrature (GGLQ) rule [17]. Arulampalam proposed the particle filter (PF) using the sequential Monte Carlo method, which is a powerful state estimation method for handling non-Gaussian noise with arbitrary non-linear models and arbitrary noise distributions [18]. However, the weights of sigma points could be negative in the unscented transformation (UT), so UKF does not behave stably in high-dimensional systems. As compared to UKF and CKF, HCKF has a very high computational burden. Particle filters suffer from particle degradation and a heavy computational burden.

Considering both filtering stability and computational demands, CKF is the more suitable choice for solving non-linear filtering problems in this work.

In practice, various unknown factors, such as environmental factors and equipment failures, can disturb navigation systems, and when data are transmitted along unreliable communication channels, the phenomenon of delayed measurements cannot be ignored [19]. The estimation accuracy is dramatically reduced when filters use delayed measurements. Handling the challenges associated with randomly delayed measurements and ensuring the accuracy of state estimation are crucial for spacecraft-relative navigation. Wang derived a Gaussian approximation (GA) filter using Gaussian approximation of the posterior predictive PDFs of states and delayed measurements [20]. Zhao adopted Bernoulli random variables (BRV) to re-represent the likelihood PDF in exponential multiplicative form and approximated the state vector with variational Bayesian methods in order to propose an improved one-step, randomly delayed KF [21]. Fei proposed a modified adaptive EKF (MAEKF) algorithm, which adopts adaptive estimation to alleviate the effects of modeling uncertainty and error [22]. Hermoso used BRV to build measurement information and proposed a randomly delayed EKF. However, these filters are all restricted to one-step or two-step delay [23,24]. Esmzad proposed a multiple-step randomly delayed CKF (MRD-CKF) that computes the likelihood function through marginalizing delay variables to mitigate the random delay and loss [25]. In the aforementioned research, however, the impact of outliers on the navigation accuracy is not considered, and this also degrades the filter significantly. Therefore, this paper extends the above-mentioned researches.

Outliers (contaminated Gaussian distribution) degrade the effectiveness of Gaussian filters. Relevant robust filters have been developed to address this problem. The robust Student's *t* filter (STF) replaces the Gaussian probability distribution in the Gaussian filtering framework with the *t*-distribution to obtain robustness [26]. Huang indicated that the STF required information regarding the degrees-of-freedom and scale matrix of the *t*-distribution, corresponding to measurement noise [27]. However, it is difficult to determine the relevant parameters of the *t*-distribution in advance. Another approach is to use generalized great likelihood estimations to revise the updating process of KF to obtain robustness [28]. The performances of robust filters based on generalized great likelihood estimations mainly depend on robust kernel functions, which restrict the anomalous measurements [29]. The most widely used robust filter adopts a Huber kernel function that combines minimum l1 and l2 norm estimation techniques [30]. Gaussian kernel function is another robust kernel function, namely, the maximum correntropy criterion (MCC). Since the weight function is smaller for the same residual, the MCC-based KF has more robustness than the Huber-based KF [31].

In this paper, we develop a non-linear, multiple-step, randomly delayed, robust filter and refer to as the multiple-step, randomly delayed, dynamic-covariance-scaling cubature Kalman filter (MRD-DCSCKF). The time update of MRD-DCSCKF is derived from the third-degree, the spherical-radial cubature rule, and the multiple-step, randomly delayed system model. The proposed filter adopts a multiple-step, randomly delayed filtering framework to weaken the impact of delayed measurements on the estimation accuracy of the filter, and it suppresses outliers with a dynamic-covariance-scaling kernel, which incorporates the advantages of the Huber kernel and Gaussian kernel.

The remaining part of the paper is organized as follows. In Section 2, some preliminaries are briefly reviewed. Section 3 presents the DCS kernel. Section 4 presents the derivation of MRD-DCSCKF. Section 5 introduces the vision-based spacecraft-relative navigation model. The simulation is reported in Section 6. Finally, conclusions are drawn in Section 7.

## 2. Preliminaries

The general discrete-time non-linear system is as follows:

$$\boldsymbol{x}_k = f(\boldsymbol{x}_{k-1}) + \boldsymbol{\varsigma}_{k-1} \tag{1}$$

$$z_k = h(x_k) + w_k \tag{2}$$

where $x_k$ is the $n$-dimensional state vector at time $t_k$. The variable $z_k$ generated from the sensor is the ideal measurement vector without delay; $\varsigma_{k-1}$ is the process noise, which satisfies zero-mean Gaussian white noises ($E\left[\varsigma_k \varsigma_l^T\right] = Q_k \delta_{kl}$); and $\delta$ indicates the Kronecker delta function. The variable $w_k$ is the measurement noise, and $f(\cdot)$ and $h(\cdot)$ denote the state function and measurement function, respectively.

### 2.1. Measurements with Multiple-Step Random Delays

The measurements obtained from Equation (2) need to proceed to the data processor (filter). However, the phenomenon of measurements transmission along unreliable communication channels, due to the occurrence of equipment failures, bandwidth limitations, environmental disturbances [32], etc., could not be ignored, as these delays result in the ideal measurements $z_k$ generated by the sensors and the actual measurements $y_k$ received by the filter becoming asynchronous. Figure 1 represents a schematic diagram of the system with multiple-step randomly delayed measurements, where the filter performs state estimation (SE) at point A and the associated measurements should reach the buffer before A. Due to the unreliability of the data transmission channel, there is one-step or multiple-step delay in the measurements received from sensor 1, where the solid line indicates that the measurements are synchronized and the dashed line indicates that the measurements are delayed. In the case of the $d$-step delay, the filter receives the actual measurements $y_k$, which could be $z_{k-i}(0 \leq i \leq d)$. Therefore, the actual measurements $y_k$ can be described as follows:

$$
\begin{aligned}
y_k &= (1 - \tau_1)z_k + \tau_1(1 - \tau_2)z_{k-1} + \tau_1\tau_2(1 - \tau_3)z_{k-2} + \\
&\quad \cdots + \left(\prod_{s=1}^{d-1} \tau_s\right)(1 - \tau_d)z_{k-d+1} + [1 - (1 - \tau_1) - \\
&\quad \tau_1(1 - \tau_2) - \cdots - \left(\prod_{s=1}^{d-1} \tau_s\right)(1 - \tau_d)]z_{k-d} \\
&= \sum_{s=0}^{d} \tau^{(s,j)} z_{k-s}
\end{aligned} \tag{3}
$$

where $\tau_0 = 1$ and $\tau_i \in [0,1]$ are independent Bernoulli random variables, which satisfy the following:

$$
\begin{aligned}
p(\tau_i = 1) &= E(\tau_i) = p_i \\
p(\tau_i = 0) &= 1 - p_i \\
E\left[(\tau_i - p_i)^2\right] &= (1 - p_i)p_i
\end{aligned} \tag{4}
$$

where $p_i$ is the expectation of $\tau_i = 1$. $\tau^{(s,j)}$ is then the following:

$$
\tau^{(s,j)} = \begin{cases} \left(\prod_{j=0}^{s} \tau_j\right)(1 - \tau_{s+1}), & 0 \leq s \leq d - 1 \\ \prod_{j=0}^{d} \tau_j, & s = d \end{cases} \tag{5}
$$

The probability of delayed measurements for $s(0 \leq s \leq d)$ step [33] is the following:

$$
\begin{aligned}
p^{(s,j)} &= \left(\prod_{j=0}^{s} p_j\right)(1 - p_{s+1}), \quad s = 0, 1, 2, \cdots, d - 1 \\
p^{(d,j)} &= \prod_{j=0}^{d} p_j
\end{aligned} \tag{6}
$$

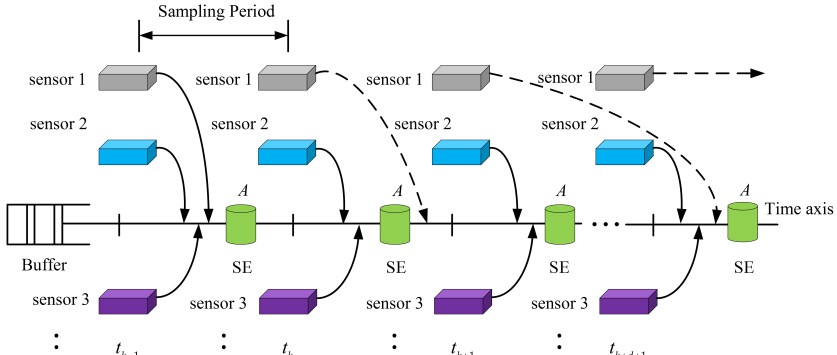

**Figure 1.** Schematic diagram of the system with multiple-step, randomly delayed measurements.

### 2.2. Non-Gaussian Noise in Measurements

A Gaussian distribution is often adopted to describe the distribution of sensor noise. However, the sensor noise does not satisfy the Gaussian assumption in practice. Furthermore, the sensors used in vision-based spacecraft-relative navigation are often perturbed by outliers, so they generally regarded as giving contaminated Gaussian distributions [30], and the probability density function (PDF) is expressed as follows:

$$p(v) = (1-\varepsilon)\frac{1}{\sqrt{2\pi}\sigma_1}\exp\left(-\frac{(v/\sigma_1)^2}{2}\right) + \varepsilon\frac{1}{\sqrt{2\pi}\sigma_2}\exp\left(-\frac{(v/\sigma_2)^2}{2}\right) \tag{7}$$

where $\varepsilon \in [0,1]$ is the perturbing parameter that denotes contamination probability. $\sigma_1$ and $\sigma_2$ are standard deviations of the individual Gaussian distributions, which are satisfied with $\|\sigma_2\| > \|\sigma_1\|$. Figure 2 presents an example that compares the Gaussian distribution and the contaminated Gaussian distribution with $\varepsilon = 0.05$, $\sigma_1 = 1$ and $\sigma_2 = 7.5\sigma_1$. Obviously, the contaminated Gaussian distribution exhibits more clutter.

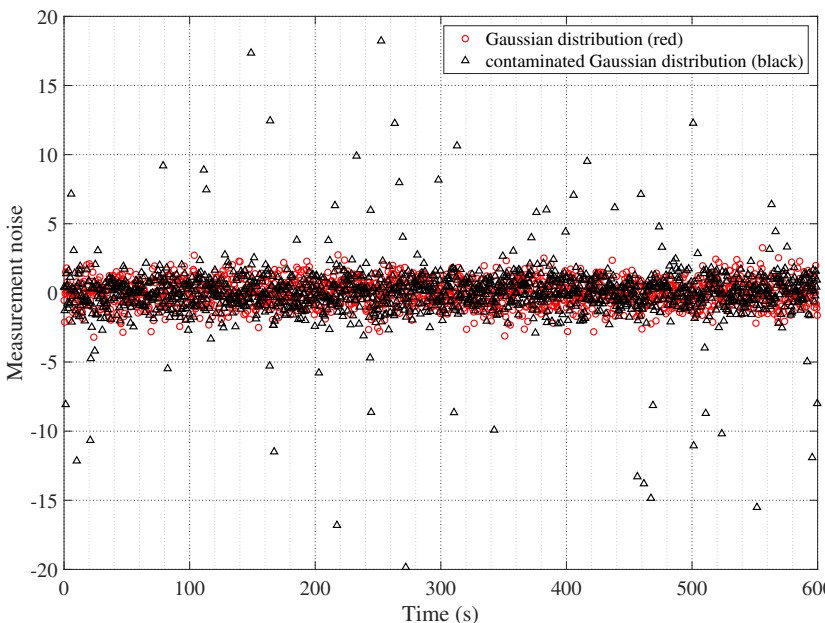

**Figure 2.** The measurement noise of the two distributions.

### 3. Brief Review of DCS Kernel

The DCS approach originates from the area of visual simultaneous localization and mapping (SLAM), and its core is a robust function, namely, the dynamic-covariance-scaling kernel. The DCS kernel enables L2 estimation to be optimal under Gaussian environment, and can completely eliminate the effect of relatively large residuals on the estimation results under non-Gaussian environment. Therefore, the DCS kernel is more robust than

the Huber kernel and can reduce the loss of measurement information under Gaussian noise, as compared to the Gaussian kernel [34].

Two highly efficient approaches, switchable constraint (SC) and dynamic covariance scaling (DCS), have been been rapidly developed in SLAM to handle outliers in images. Agarwal [35] has derived the specific form of the cost function of SC, as follows:

$$\rho(\xi) = s^2\xi^2 + \eta(1 - s)^2 \tag{8}$$

where $s$ is the switchable variable, $\eta > 0$ is the kernel width, and $\xi$ is the residual error.

When the function $\rho(\cdot)$ is continuous and bounded, the derivative of Equation (8) concerning $s$ should satisfy the following conditions:

$$\frac{\partial \rho(\xi)}{\partial s} = 2s\xi^2 - 2\eta(1 - s) = 0 \tag{9}$$

$$s = \frac{\eta}{\eta + \xi^2} \tag{10}$$

By bringing Equation (10) into Equation (8), the specific form of the cost function of SC could be derived, as follows:

$$\rho(\xi) = \left(\frac{\eta}{\eta + \xi^2}\right)^2 \xi^2 + \eta\left(1 - \frac{\eta}{\eta + \xi^2}\right)^2 = \frac{\eta\xi^2}{\eta + \xi^2} \tag{11}$$

The limit value of $\rho(\cdot)$ at $\pm\infty$ is determined by the following:

$$\lim_{\xi \to \pm\infty} \rho(\xi) = \lim_{\xi \to \pm\infty} \frac{\eta}{\eta/\xi^2 + 1} = \eta \tag{12}$$

According to Equation (12), $\eta$ is the upper bound of $\rho(\cdot)$. We could determine the range of $s$.

$$s \in \left[0, \frac{2\eta}{\eta + \xi^2}\right] \tag{13}$$

Sünderhauf [36] suggested that the range of values for $s$ is [0,1], and when this was combined with Equation (13), it resulted in the following:

$$s = \min\left(1, \frac{2\eta}{\eta + \xi^2}\right) \tag{14}$$

Agarwal [35] suggested that $s^2$ could be used as a form of weight function for M-estimation, and then the DCS weight function could be calculated, as follows:

$$\psi_D(\xi) = s^2 = \min\left(1, \frac{4\eta^2}{(\eta + \xi^2)^2}\right) = \begin{cases} 1, & \text{for } \xi^2 < \eta \\ \frac{4\eta^2}{(\eta+\xi^2)^2}, & \text{for } \xi^2 \geq \eta \end{cases} \tag{15}$$

By integrating Equation (15), the cost function of DCS is as follows:

$$\rho_D(\xi) = \int \xi\psi_D(\xi)d\xi = \begin{cases} \frac{\xi^2}{2}, & \text{for } \xi^2 < \eta \\ \frac{\eta(3\xi^2 - \eta)}{2(\xi^2 + \eta)}, & \text{for } \xi^2 \geq \eta \end{cases} \tag{16}$$

The cost function and the weight function of a DCS kernel are shown in Figure 3.

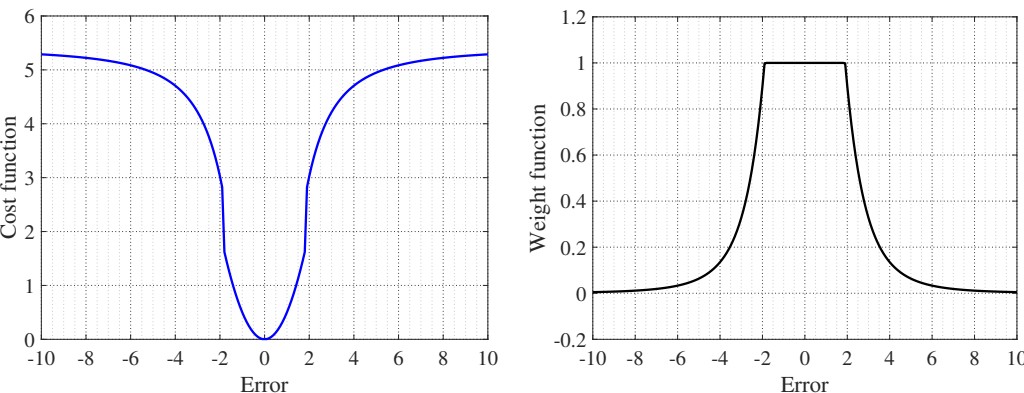

**Figure 3.** Cost and weight functions of a DCS kernel.

## 4. Multiple-Step, Randomly Delayed, Robust Cubature Kalman Filter

### 4.1. State-Augmentation

According to Equations (2) and (3), the actual measurement $y_k$ received by the filter is a mixture of $\{z_{k-s}\}_{s=0}^{d}$ . The derivation of $y_k$ needs the state vectors $x_k, x_{k-1}, \ldots, x_{k-d}$ , which are then joined together. The augmented state vector is as follows:

$$X_k = \begin{bmatrix} x_k \\ x_{k-1} \\ \vdots \\ x_{k-d} \end{bmatrix}_{n_a \times 1} \tag{17}$$

where $X_k$ denotes the $n_a = (d+1)n$-dimensional augmented state.

The augmentation system is expressed by the following:

$$X_k = F(X_{k-1}) + C\varsigma_{k-1} \tag{18}$$

$$y_k = z_{k-s} = h(D_s X_k) + w_{k-1}, \; s = 0, 1, \ldots, d \tag{19}$$

where $F(X_{k-1})$ , $C$ , and $D_s X_k$ are then computed by the following:

$$F(X_{k-1}) = \left[ f^T(x_{k-1}), x_{k-1}^T, \ldots, x_{k-d}^T \right]^T \tag{20}$$

$$C = [I_{n \times n}, 0, \ldots, 0]^T \tag{21}$$

$$D_s X_k = x_{k-s}, \; s = 0, 1, 2, \ldots, d \tag{22}$$

### 4.2. Prediction

Based on Equation (17), the state estimation and error covariance are expressed by the following:

$$\hat{X}_{k-1/k-1} = \begin{bmatrix} \hat{x}_{k-1/k-1} \\ \vdots \\ \hat{x}_{k-d/k-1} \\ \hat{x}_{k-d-1/k-1} \end{bmatrix} \tag{23}$$

$$P_{k-1/k-1} = \begin{bmatrix} P_{k-1,k-1/k-1}^{xx} & \cdots & P_{k-1,k-d/k-1}^{xx} & P_{k-1,k-d-1/k-1}^{xx} \\ \vdots & \ddots & \vdots & \vdots \\ P_{k-d,k-1/k-1}^{xx} & \cdots & P_{k-d,k-d/k-1}^{xx} & P_{k-d,k-d-1/k-1}^{xx} \\ P_{k-d-1,k-1/k-1}^{xx} & \cdots & P_{k-d-1,k-d/k-1}^{xx} & P_{k-d-1,k-d-1/k-1}^{xx} \end{bmatrix} \tag{24}$$

The predicted state $\hat{X}_{k/k-1}$ and corresponding covariance $P_{k/k-1}$ are determined by the following:

$$\hat{X}_{k/k-1} = \begin{bmatrix} \hat{x}_{k/k-1} \\ \hat{x}_{k-1/k-1} \\ \vdots \\ \hat{x}_{k-d/k-1} \end{bmatrix} \tag{25}$$

$$P_{k/k-1} = \begin{bmatrix} P_{k,k/k-1}^{xx} & P_{k,k-1/k-1}^{xx} & \cdots & P_{k,k-d/k-1}^{xx} \\ P_{k-1,k/k-1}^{xx} & P_{k-1,k-1/k-1}^{xx} & \cdots & P_{k-1,k-d/k-1}^{xx} \\ \vdots & \vdots & \ddots & \vdots \\ P_{k-d,k/k-1}^{xx} & P_{k-d,k-1/k-1}^{xx} & \cdots & P_{k-d,k-d/k-1}^{xx} \end{bmatrix} \tag{26}$$

According to Appendix A.1, the cubature points are generated according to $\hat{X}_{k-1/k-1}$ and $P_{k-1/k-1}$ as follows:

$$\chi_{i,k-1|k-1} = \text{Trans}\left[\hat{X}_{k-1/k-1}, P_{k-1/k-1}\right] \tag{27}$$

The predicted state and corresponding covariance matrix are expressed as Equations (28)–(31).

$$\hat{x}_{k|k-1} = \sum_{i=1}^{2n_a} w_i f\left(\chi_{i,k-1|k-1}\right) \tag{28}$$

$$P_{k,k/k-1}^{xx} = \sum_{i=1}^{2n_a} w_i \left(f\left(\chi_{i,k-1|k-1}\right) - \hat{x}_{k|k-1}\right)\left(f\left(\chi_{i,k-1|k-1}\right) - \hat{x}_{k|k-1}\right)^T + Q_{k-1} \tag{29}$$

$$P_{k,k-s/k-1}^{xx} = \sum_{i=1}^{2n_a} w_i \left(f\left(\chi_{i,k-1|k-1}\right) - \hat{x}_{k|k-1}\right)\left(\chi_{i,k-s|k-1} - \hat{x}_{k-s|k-1}\right)^T, s = 1, 2, \cdots, d \tag{30}$$

$$P_{k-s,k/k-1}^{xx} = \left(P_{k,k-s/k-1}^{xx}\right)^T, \ s = 1, 2, \cdots, d \tag{31}$$

*4.3. Update*

The cubature points generated with $\hat{X}_{k/k-1}$ and $P_{k/k-1}$ is calculated as follows:

$$\chi_{i,k|k-1} = \text{Trans}\left[\hat{X}_{k/k-1}, P_{k/k-1}\right] \tag{32}$$

We calculate the measurement prediction $\hat{y}_{k/k-1}^s$, covariance $P_{k/k-1}^{yy,s}$, and cross-covariance $P_{k/k-1}^{Xy,s}$ at the $s$-th $(0 \leq s \leq d)$ step delay as shown in Equations (33)–(35).

$$\hat{y}_{k/k-1}^s = \sum_{i=1}^{2n_a} w_i h\left(D_s \chi_{i,k|k-1}\right) \tag{33}$$

$$P_{k/k-1}^{yy,s} = \sum_{i=1}^{2n_a} w_i \left(h\left(D_s \chi_{i,k|k-1}\right) - \hat{y}_{k/k-1}^s\right)\left(h\left(D_s \chi_{i,k|k-1}\right) - \hat{y}_{k/k-1}^s\right)^T \tag{34}$$

$$P_{k/k-1}^{Xy,s} = \sum_{i=1}^{2n_a} w_i \left( \chi_{i,k|k-1} - \hat{X}_{k|k-1} \right) \left( h \left( D_s \chi_{i,k|k-1} \right) - \hat{y}_{k/k-1}^s \right)^T \tag{35}$$

The sub-update results of the multiple-step randomly delayed cubature Kalman filter at the *s*-th step are provided in Appendix A.2. In this study, the generalized maximum likelihood approach is adopted to complete the sub-update of MRD-DCSCKF for the purpose of suppressing outliers. The robust update calculation process is denoted by the function $\left[ M, \hat{z} \right] = \text{Robust\_update} \left[ y_k, h(\cdot), P_{k/k-1}, P^{xy}, \hat{x}_{k|k-1}, R \right]$, according to Appendix A.3. The corresponding quantities generated at the *s*-th $(0 \le s \le d)$ step delay is expressed as follows:

$$\left[ M_k^s, \hat{z}_k^s \right] = \text{Robust\_update} \left[ y_k, h(\cdot), P_{k/k-1}, P_{k/k-1}^{Xy,s}, \hat{X}_{k|k-1}, R_{k-s} \right] \tag{36}$$

It is well known that the robustness of M-estimation in a non-Gaussian noise environment mainly depends on the robust kernel function, and this paper adopts the DCS robust kernel function introduced in Section 3. The residuals and weight matrix at *s*-th $(0 \le s \le d)$ step delay are as follows:

$$\xi_k^s = M_k^s \hat{X}_k^s - \hat{z}_k^s \tag{37}$$

$$\mathbf{\Psi}_D^s = diag \left[ \psi_D \left( \xi_k^{l,s} \right) \right] \tag{38}$$

where $\xi_k^{l,s}$ is the *l*-th element of residuals $\xi_k^s$. In addition, $\psi_D(\cdot)$ is the DCS weight function expressed in Equation (15), and $\mathbf{\Psi}$ is the weight matrix, which denotes the proportion of residuals accounted for. The initial value of the iteration of $\hat{X}_k^s$ could be chosen based on Appendix A.2. After *j* iterations of Equations (37)–(38), we could obtain sub-update results at the *s*-th step delay.

$$\hat{X}_{k/k}^{(j+1),s} = \left( (M_k^s)^T \mathbf{\Psi}_D^{j,s} M_k^s \right)^{-1} (M_k^s)^T \mathbf{\Psi}_D^{j,s} \hat{z}_k^s \tag{39}$$

$$P_{k|k}^{(j+1),s} = \left( (M_k^s)^T \mathbf{\Psi}_D^{j,s} M_k^s \right)^{-1} \tag{40}$$

The state estimate and corresponding covariance with sub-update results can be computed as:

$$\hat{X}_{k/k} = \sum_{s=0}^{d} \mu_k^s \hat{X}_{k/k}^s \tag{41}$$

$$P_{k/k} = \sum_{s=0}^{d} \mu_k^s P_{k/k}^s \tag{42}$$

where $\mu_k^s$ is the weight of the sub-update results, according to Equation (43), which is computed based on delay probabilities $p^s$, measurement prediction $\hat{y}_{k/k-1}^s$, and covariance $P_{k/k-1}^{yy,s}$ [25].

$$\mu_k^s = \frac{p^s N \left( y_k; \hat{y}_{k/k-1}^s, P_{k/k-1}^{yy,s} + R_{k-s} \right)}{\sum\limits_{i=0}^{d} \left[ p^i N \left( y_k; \hat{y}_{k/k-1}^i, P_{k/k-1}^{yy,i} + R_{k-i} \right) \right]} \tag{43}$$

In delayed system, the data received by the filter may be the measurements without delay (s = 0) or delayed measurements $(1 \le s \le d)$. The state-augmentation approach can effectively use the previous states of the system, namely, the state information, such as (s = 0,1...d), is sufficiently exploited in MRD-DCSCKF. Through calculating the weights of sub-updates to obtain the posterior PDF of the Gaussian mixture, the MRD-DCSCKF

algorithm can obtain accurate information in the augmented states and improve the state estimation accuracy. Finally, to better illustrate the computational procedures of MRD-DCSCKF, the primary calculations of the proposed filter are outlined in Figure 4, where the non-occurrence delay $(s = 0)$ and the occurrence delay $(1 \leq s \leq d)$ are drawn separately to distinguish them from each other.

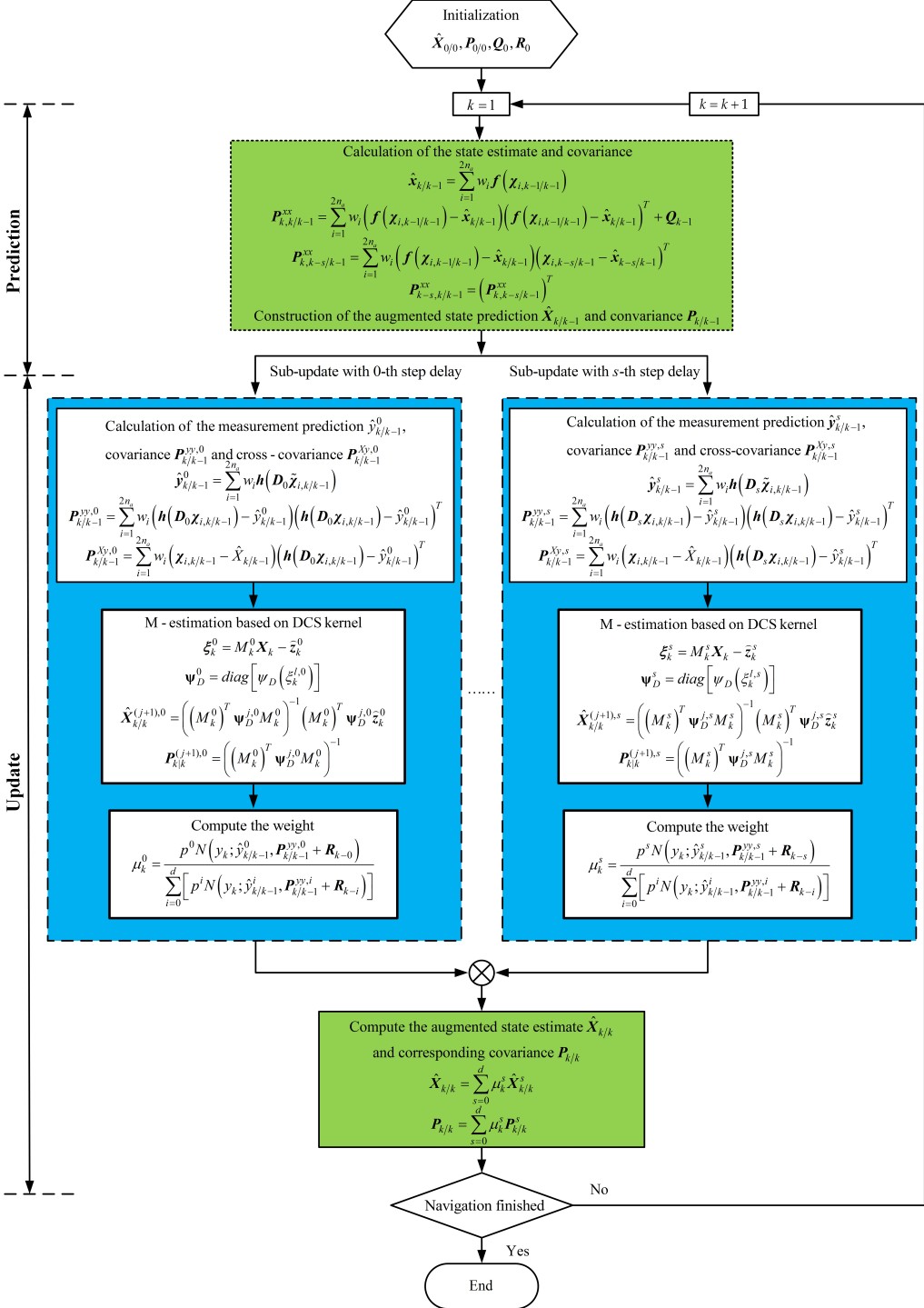

**Figure 4.** Flowchart of the MRD-DCSCKF.

## 5. Spacecraft-Relative Navigation Model

### 5.1. Reference Frames

To describe the spacecraft's relative motion and sensor measurement model, three related coordinate frames are defined in this section, as shown in Figure 5.

1.  Earth-centered inertial (ECI) frame ($O_I - X_I Y_I Z_I$): The origin $O_I$ is located at the center of Earth, the $X_I$-axis points to the vernal equinox, the $Z_I$-axis points to the North Pole, and the $Y_I$-axis forms a right-handed system with the $X_I$-axis and the $Z_I$-axis.
2.  Local–vertical–local–horizontal (LVLH) frame ($O_L - X_L Y_L Z_L$): The origin $O_L$ is located at the mass center of the chief spacecraft, the $X_L$-axis points from the center of Earth to the center of the chief spacecraft, the $Z_L$-axis points in the same direction as the orbital angular velocity, and the $Y_L$-axis forms a right-handed system with the $X_L$-axis and the $Z_L$-axis.
3.  Spacecraft body coordinate frame ($O_b - X_b Y_b Z_b$): The chief body and deputy body are denoted as $O_c - X_c Y_c Z_c$ and $O_d - X_d Y_d Z_d$, respectively. The $O_b - X_b Y_b Z_b$ is fixed to the spacecraft and its origin $O_b$ is located at the mass center of the spacecraft. The three axes of $O_b - X_b Y_b Z_b$, a right-handed system, coincide with the inertial axes of spacecraft. When $O_c - X_c Y_c Z_c$ is coincident with $O_L - X_L Y_L Z_L$, the $X_c$-axis points outward radially along the orbit (yaw axis), the $Y_c$-axis points toward the direction of flight (roll axis), and the $Z_c$-axis forms a right-handed system with the $X_c$-axis and the $Y_c$-axis (pitch axis).

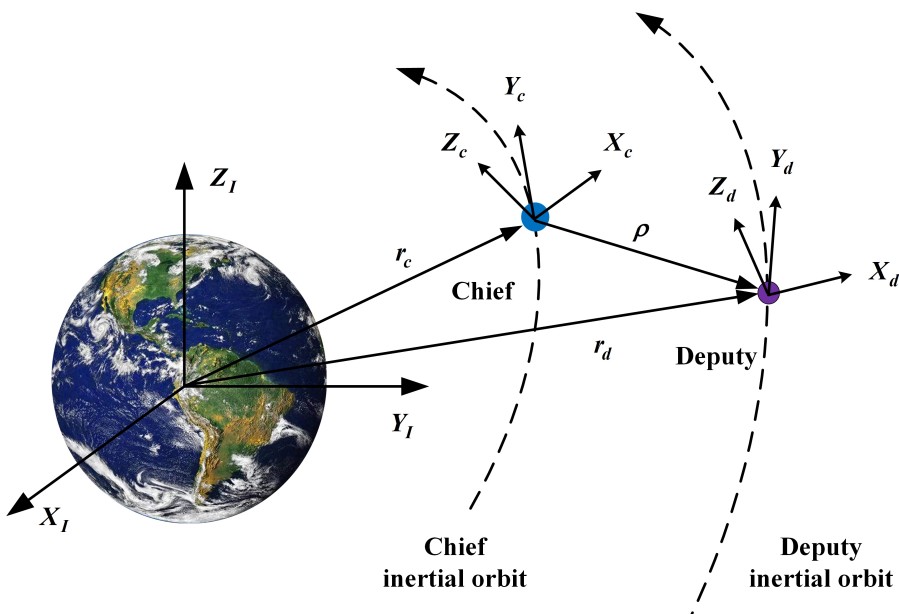

**Figure 5.** Definition of related coordinate frames.

### 5.2. Relative Dynamics

The chief and deputy spacecraft move along in Earth-centered orbits, as shown in Figure 5. The relative position vector of the deputy spacecraft with respect to the chief spacecraft is $\boldsymbol{\rho} = \boldsymbol{r}_d - \boldsymbol{r}_c = [x, y, z]^T$ in the LVLH frame based on the chief spacecraft. The nonlinear dynamics of the relative motion between them are described with Tschauner–Hempel (T-H) equations [37]. Then, the relative dynamics are calculated using the following equations:

$$\ddot{x} = 2\dot{v}\dot{y} + \ddot{v}y + \dot{v}^2 x - \frac{\mu(r_c + x)}{\left[(r_c + x)^2 + y^2 + z^2\right]^{3/2}} + \frac{\mu}{r_c^2} \tag{44}$$

$$\ddot{y} = -2\dot{v}\dot{x} - \ddot{v}x + \dot{v}^2 y - \frac{\mu y}{\left[ (r_c + x)^2 + y^2 + z^2 \right]^{3/2}} \tag{45}$$

$$\ddot{z} = -\frac{\mu z}{\left[ (r_c + x)^2 + y^2 + z^2 \right]^{3/2}} \tag{46}$$

where $\mu$ is the gravitational parameter of Earth, $v$ denotes the true anomaly, and $r_c$ is the orbit's radius. They are described by $\ddot{v} = \frac{-2\dot{r}_c\dot{v}}{r_c}$ and $\ddot{r}_c = r_c\dot{v}^2 - \frac{\mu}{r_c^2}$ , respectively.

### 5.3. Relative Attitude Kinematics

Since the attitude kinematics described by quaternions have a linear form, it is more convenient to calculate the spacecraft attitude with quaternions [38]. The quaternion is defined by the following:

$$q \equiv \begin{bmatrix} \varrho \\ q_4 \end{bmatrix} \tag{47}$$

with

$$\varrho \equiv [q_1, q_2, q_3]^T = e \sin(\vartheta/2) \tag{48}$$

$$q_4 = \cos(\vartheta/2) \tag{49}$$

where $e$ and $\vartheta$ denote the unit Euler axis and the rotation angle, respectively. Quaternions satisfy $q^T q = \|q\|^2 = 1$ . The relationship between quaternions and the attitude matrix is as follows:

$$A(q) = \Xi^T(q)\psi(q) \tag{50}$$

with

$$\Xi(q) \equiv \begin{bmatrix} q_4 I_{3\times3} + [\varrho \times] \\ -\varrho^T \end{bmatrix} \tag{51}$$

$$\psi(q) \equiv \begin{bmatrix} q_4 I_{3\times3} - [\varrho \times] \\ -\varrho^T \end{bmatrix} \tag{52}$$

where $[\varrho \times]$ is the cross-product matrix, provided by the following:

$$[\varrho \times] \equiv \begin{bmatrix} 0 & -q_3 & q_2 \\ q_3 & 0 & -q_1 \\ -q_2 & q_1 & 0 \end{bmatrix} \tag{53}$$

Successive rotations could be converted into quaternion multiplication, as follows:

$$A(q)A(q') = A(q \otimes q') \tag{54}$$

with

$$q \otimes q' = [\psi(q) \quad q]q' = [\Xi(q') \quad q']q \tag{55}$$

The quaternion kinematics equation could be written as the following:

$$\dot{q} = \frac{1}{2}\Xi(q)\omega = \frac{1}{2}\Omega(\omega)q \tag{56}$$

where $\omega = [\omega_1, \omega_2, \omega_3]^T$ indicates the angular rate vector and

$$\Omega(\omega) \equiv \begin{bmatrix} -[\omega \times] & \omega \\ -\omega^T & 0 \end{bmatrix} \tag{57}$$

The error of quaternion is defined as $\delta\boldsymbol{q} = \left[\delta\boldsymbol{\varrho}^T, \delta q_4\right]^T$. To ensure that the quaternion-based results follow the unit constraint, generalized Rodrigues parameters (GRPs) [10] are introduced to represent the pose error, which is defined as follows:

$$\delta\boldsymbol{p} \equiv f\frac{\delta\boldsymbol{\varrho}}{(a + \delta q_4)} \tag{58}$$

where $\|\delta\boldsymbol{p}\|$ is equal to the error-angle for small errors, $f$ is a scale factor that is commonly taken as $f = 2(a + 1)$, and $a$ is a parameter from 0 to 1. The inverse transformation from $\delta\boldsymbol{p}$ to $\delta\boldsymbol{q}$ is determined as follows:

$$\delta q_4 = \frac{-a\|\delta\boldsymbol{p}\|^2 + f\sqrt{f^2 + (1 - a^2)\|\delta\boldsymbol{p}\|^2}}{f^2 + \|\delta\boldsymbol{p}\|^2} \tag{59}$$

$$\delta\boldsymbol{\varrho} = f^{-1}(a + \delta q_4)\delta\boldsymbol{p} \tag{60}$$

*5.4. Measurement Model*

Several optical beacons of known number and position are mounted on the chief spacecraft. The deputy spacecraft is equipped with PSD, and this paper does not assume that chief body $(O_c - X_c Y_c Z_c)$ and its LVLH frame $(O_L - X_L Y_L Z_L)$ are consistent. When the chief spacecraft and deputy spacecraft are close to each other, e.g., distances ranging from several meters to several hundred meters [39], the details of each part of chief spacecraft could be displayed on the pixel plane [40]. Based on the measurements from the cameras, it is possible to estimate the relative position and velocity between them and also to estimate their relative attitude. The VISNAV system is illustrated in Figure 6, where PSD is denoted with a cube, and this paper assumes that the coordinate system of PSD overlaps with deputy body $O_d - X_d Y_d Z_d$. The $i$-th measurement line of sight (LOS) vector is defined by the following:

$$\tilde{\boldsymbol{b}}_i = A\left(\boldsymbol{q}_{d/L}\right)\boldsymbol{r}_i + \boldsymbol{w}_i \tag{61}$$

with

$$\boldsymbol{r}_i = \frac{\boldsymbol{X}'_i - \boldsymbol{\rho}}{\|\boldsymbol{X}'_i - \boldsymbol{\rho}\|} = \frac{[X'_i - x, Y'_i - y, Z'_i - z]^T}{\sqrt{(X'_i - x)^2 + (Y'_i - y)^2 + (Z'_i - z)^2}} \tag{62}$$

where $\tilde{\boldsymbol{b}}_i$ is the measurement of the $i$-th beacon in $O_d - X_d Y_d Z_d$. $(x, y, z)$ indicates the position of deputy spacecraft in $O_L - X_L Y_L Z_L$. $A\left(\boldsymbol{q}_{d/L}\right)$ denotes the rotation matrix from $O_L - X_L Y_L Z_L$ to $O_d - X_d Y_d Z_d$. In addition, $\boldsymbol{X}'_i = [A(\boldsymbol{q}_{c/L})]^T \boldsymbol{X}_i$ is the coordinate of beacon in $O_L - X_L Y_L Z_L$, and $\boldsymbol{X}_i \equiv (X_i, Y_i, Z_i)$ is the $i$-th beacon's known position in $O_c - X_c Y_c Z_c$.

Therefore, with N optical beacons providing N lines of sight, the optical camera measurement equation is written as follows:

$$\boldsymbol{z}_k = \begin{bmatrix} \tilde{\boldsymbol{b}}_1 \\ \tilde{\boldsymbol{b}}_2 \\ \vdots \\ \tilde{\boldsymbol{b}}_N \end{bmatrix}_k = \begin{bmatrix} A\left(\boldsymbol{q}_{d/L}\right)\boldsymbol{r}_1 \\ A\left(\boldsymbol{q}_{d/L}\right)\boldsymbol{r}_2 \\ \vdots \\ A\left(\boldsymbol{q}_{d/L}\right)\boldsymbol{r}_N \end{bmatrix}_k + \begin{bmatrix} \boldsymbol{w}_1 \\ \boldsymbol{w}_2 \\ \vdots \\ \boldsymbol{w}_N \end{bmatrix}_k \tag{63}$$

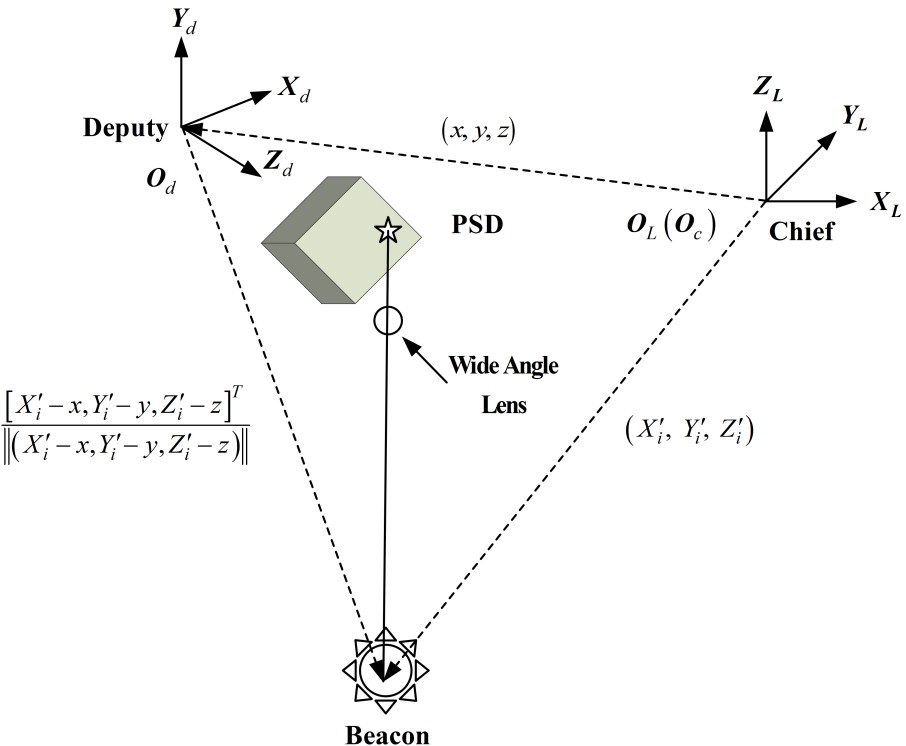

**Figure 6.** Vision-based navigation system.

*5.5. Gyro Measurement Model*

The classical model of the angular rate of the gyro output [41] is expressed by the following:

$$\begin{cases} \tilde{\omega} = \omega + \beta + \eta_v \\ \dot{\beta} = \eta_u \end{cases} \tag{64}$$

where $\tilde{\omega}$ is the actual angular rate, and $\omega$ is the true inertial angular rate. The variable $\beta$ is the gyro drift, and $\eta_v$ and $\eta_u$ are Gaussian white-noise, which satisfy

$$\begin{cases} E\{\eta_v(t)\eta_v^T(\tau)\} = \sigma_v^2\delta(t-\tau)I_{3\times3} \\ E\{\eta_u(t)\eta_u^T(\tau)\} = \sigma_u^2\delta(t-\tau)I_{3\times3} \end{cases} \tag{65}$$

where $\sigma_v$ and $\sigma_u$ indicate the angle random walk and angular rate random walk, respectively.

The discrete recursive form of Equation (64) is defined as follows:

$$\begin{cases} \tilde{\omega}_{k+1} = \omega_{k+1} + \frac{1}{2}(\beta_{k+1} + \beta_k) + \left(\frac{\sigma_v^2}{\Delta t} + \frac{1}{12}\sigma_u^2\Delta t\right)^{1/2}N_v \\ \beta_{k+1} = \beta_k + \sigma_u\Delta t^{1/2}N_u \end{cases} \tag{66}$$

where $\Delta t$ is the discrete sampling period. The variables $N_v$ and $N_u$ are the zero-mean Gaussian white-noise.

## 6. Numerical Simulations

To demonstrate the superiority of MRD-DCSCKF for handling multiple-step, randomly delayed measurements and suppressing outliers, we designed simulations to compare the properties of the proposed filter with CKF [15], the one-step randomly delayed CKF (ORD-CKF) [20], and the multiple-step randomly delayed CKF (MRD-CKF) [25].

*6.1. Experimental Scenario Settings*

The installation positions of the optical beacons under the chief spacecraft body coordinate frame are shown in Table 1. Table 2 lists the initial orbital elements of the chief spacecraft. Table 3 summarizes the relevant parameters of the simulation. This paper

assumes that the chief spacecraft gyro drift is relatively small, or it has been corrected by autonomous navigation. Therefore, chief spacecraft gyro drift $\beta_{c0} = 0$ is adopted. The nominal trajectory of the chief spacecraft is depicted in Figure 7.

**Table 1.** Installation positions of beacons.

| Beacon No. | x (m) | y (m) | z (m) |
|:---:|:---:|:---:|:---:|
| 1 | 0.5 | 0.5 | 0.0 |
| 2 | −0.5 | 0.5 | 0.0 |
| 3 | 0.5 | −0.5 | 0.0 |
| 4 | −0.5 | −0.5 | 0.0 |
| 5 | 0.2 | 0.5 | 0.1 |
| 6 | 0.1 | 0.2 | −0.1 |

**Table 2.** Initial orbital elements of the chief spacecraft.

| Orbital Elements | Corresponding Value |
|:---:|:---:|
| Semi-major axis $a$ | 26,555.137 km |
| Eccentricity $e$ | 0.7395 |
| Orbit inclination $i$ | 63.465° |
| Argument of perigee $\omega$ | 274.163° |
| Right ascension of the ascending node $\Omega$ | 115.024° |
| True anomaly $v$ | 23.612° |

**Table 3.** Simulation parameters.

| Parameter | Corresponding Value |
|:---:|:---:|
| Number of Monte Carlo simulations | 100 |
| Discrete sampling period | 0.1 s |
| The update interval of camera | 0.2 s |
| Simulation time | 600 s |
| Perturbing parameter | 0.05 |
| Tuning parameters of kernel | 5 |
| Number of delay steps | 3 |
| Delay probability | 0.1 |
| Delay probability for each step | $p_0 = 0.9, p_1 = 0.09, p_2 = 0.009, p_3 = 0.001$ |
| Initial relative position | $\boldsymbol{\rho}_0 = [0, -27.444, 0]^T \,(\mathrm{m})$ |
| Initial relative velocity | $\boldsymbol{v}_0 = \left[0, -6.340 \times 10^{-3}, 0\right]^T (\mathrm{m/s})$ |
| Initial attitude quaternion of chief spacecraft | $\boldsymbol{q}_{c/L} = [0.0086, 0.0086, 0.0086, 0.9999]^T$ |
| Initial relative attitude quaternion | $\boldsymbol{q}_{d/c} = [0.0433, 0.0142, 0.0256, 0.9986]^T$ |
| Initial generalized Rodrigues parameters | $\delta \boldsymbol{p}_0 = [0, 0, 0]^T$ |
| Chief spacecraft angular velocity | $\boldsymbol{\omega}_{c,0} = [0.0013, 0.0013, 0.0013]^T \,(\mathrm{rad/s})$ |
| Deputy spacecraft angular velocity | $\boldsymbol{\omega}_{d,0} = [0.0020, 0.0020, 0.0020]^T \,(\mathrm{rad/s})$ |
| Gyro drift | $\boldsymbol{\beta}_{d0} = [1, 1, 1]^T \,(°/\mathrm{h})$ |
| Angle random walk | $\sigma_u = \sqrt{10} \times 10^{-8} \,(\mathrm{rad/s^{1/2}})$ |
| Angular rate random walk | $\sigma_v = \sqrt{10} \times 10^{-10} \,(\mathrm{rad/s^{3/2}})$ |
| Power spectral density of perturbation acceleration | $q_\omega = \sqrt{10} \times 10^{-6} \left((\mathrm{m/s^2})^2 / \mathrm{Hz}\right)$ |
| Process noise covariance matrix | $\boldsymbol{Q}_k = diag\left[\sigma_u^2 \boldsymbol{I}_{3\times3}, \sigma_v^2 \boldsymbol{I}_{3\times3}, 0_{3\times3}, q_\omega^2 \boldsymbol{I}_{3\times3}\right]$ |
| Initial state covariance matrix | $\boldsymbol{P}_0 = diag\left[(1°)^2 \boldsymbol{I}_{3\times3}, (1°/\mathrm{h})^2 \boldsymbol{I}_{3\times3}, (1\mathrm{m})^2 \boldsymbol{I}_{3\times3}, (1\mathrm{m/s})^2 \boldsymbol{I}_{3\times3}\right]$ |
| Covariance matrix of measurement noise | $\boldsymbol{R}_k = (1.8'')^2 \boldsymbol{I}_{12\times12}$ |
| Covariance matrix of contaminated measurement noise | $\boldsymbol{R}_{con} = 7.5 \boldsymbol{R}_k$ |
| Initial state vector true value | $\boldsymbol{x}_0 = \left[\delta \boldsymbol{p}_0^T, \boldsymbol{\beta}_{d0}^T, \boldsymbol{\rho}_0^T, \boldsymbol{v}_0^T\right]^T$ |
| Initial state vector estimate | $\hat{\boldsymbol{x}}_0 = \boldsymbol{x}_0 + \sqrt{\boldsymbol{P}_0} \cdot \boldsymbol{N}_{12\times1}$ |

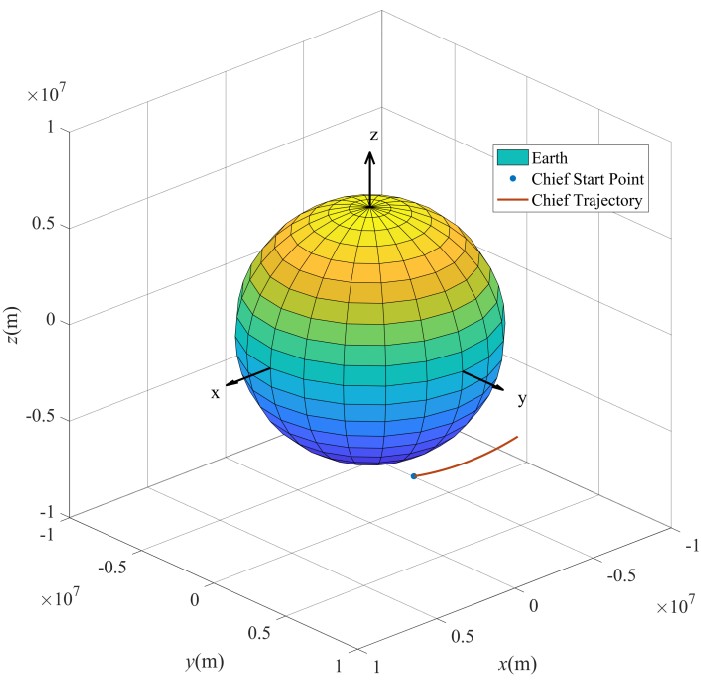

**Figure 7.** The nominal trajectory of the chief spacecraft.

### 6.2. Simulation Results and Analyses

Monte Carlo simulations were conducted to evaluate the effectiveness of several filters for estimation. Then, the root-mean-square error (*RMSE*) and the average RMSE (*ARMSE*) of the state estimations were calculated, as follows:

$$RMSE_k(i) = \sqrt{\frac{1}{M} \sum_{m=1}^{M} \left( \hat{x}_k^m(i) - x_k^m(i) \right)^2} \tag{67}$$

$$ARMSE(i) = \frac{1}{k_n - k_0} \sum_{k=k_0}^{k_n} RMSE_k(i) \tag{68}$$

where $M$ denotes the number of Monte Carlo simulations. $k$ is the $k$-th instant, and $\hat{x}_k^m$ and $x_k^m$ are the estimation and the true value, respectively.

Figures 8–11 compare the performances of the CKF, ORD-CKF, MRD-CKF, and MRD-DCSCKF. The simulation results show that CKF had the worst estimation. The performance of ORD-CKF was between CKF and MRD-CKF because ORD-CKF could only handle one-step randomly delayed measurements. The performance of MRD-CKF was significantly better than CKF and ORD-CKF because it could handle multiple-step randomly delayed measurements. There is no doubt that MRD-DCSCKF had the best performance among all filters. MRD-DCSCKF adopts the multiple-step randomly delayed filtering framework in combination with the DCS kernel function, which enables its ability to address both delayed measurements and outliers.

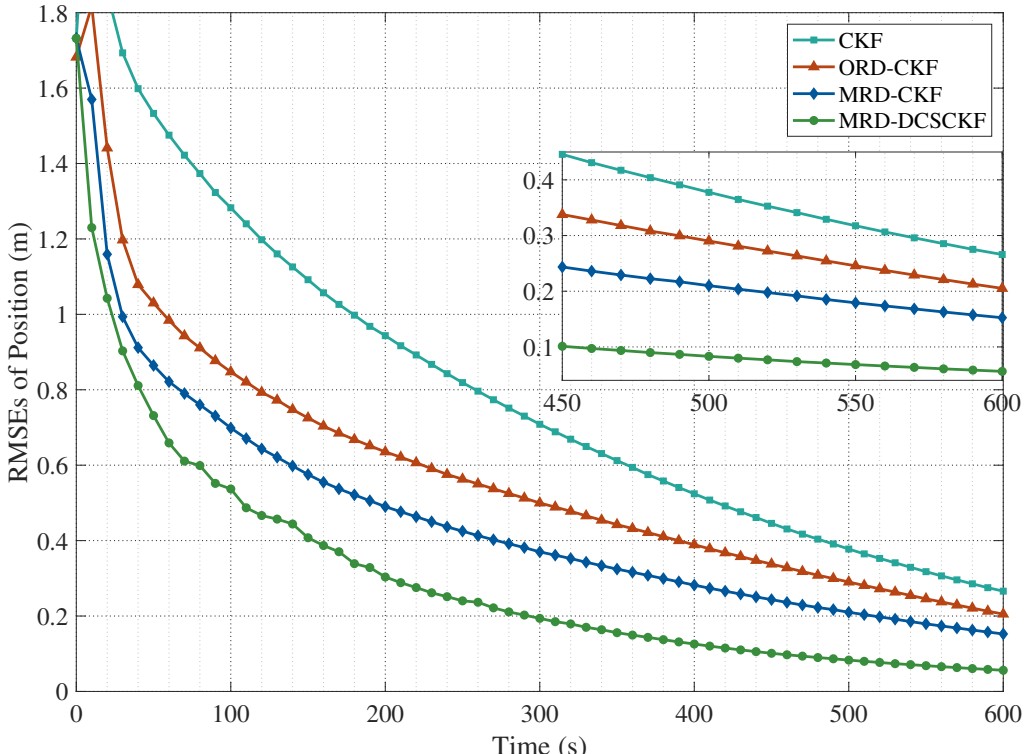

**Figure 8.** RMSEs of the position with CKF, ORD-CKF, MRD-CKF, and MRD-DCSCKF.

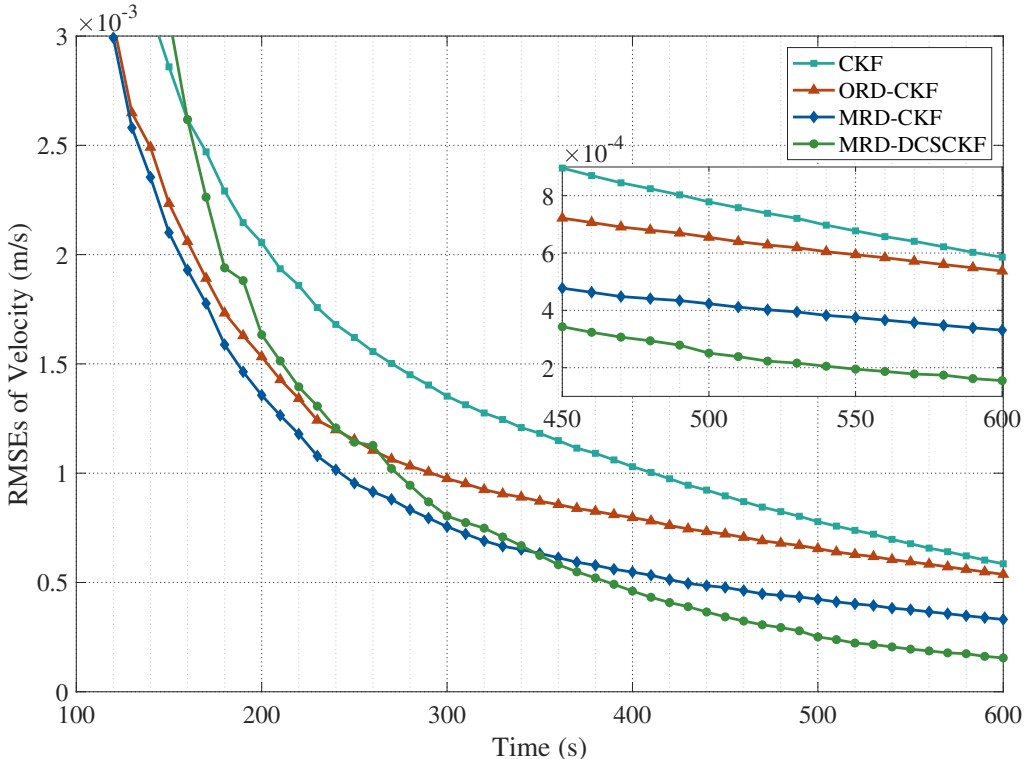

**Figure 9.** RMSEs of the velocity with CKF, ORD-CKF, MRD-CKF, and MRD-DCSCKF.

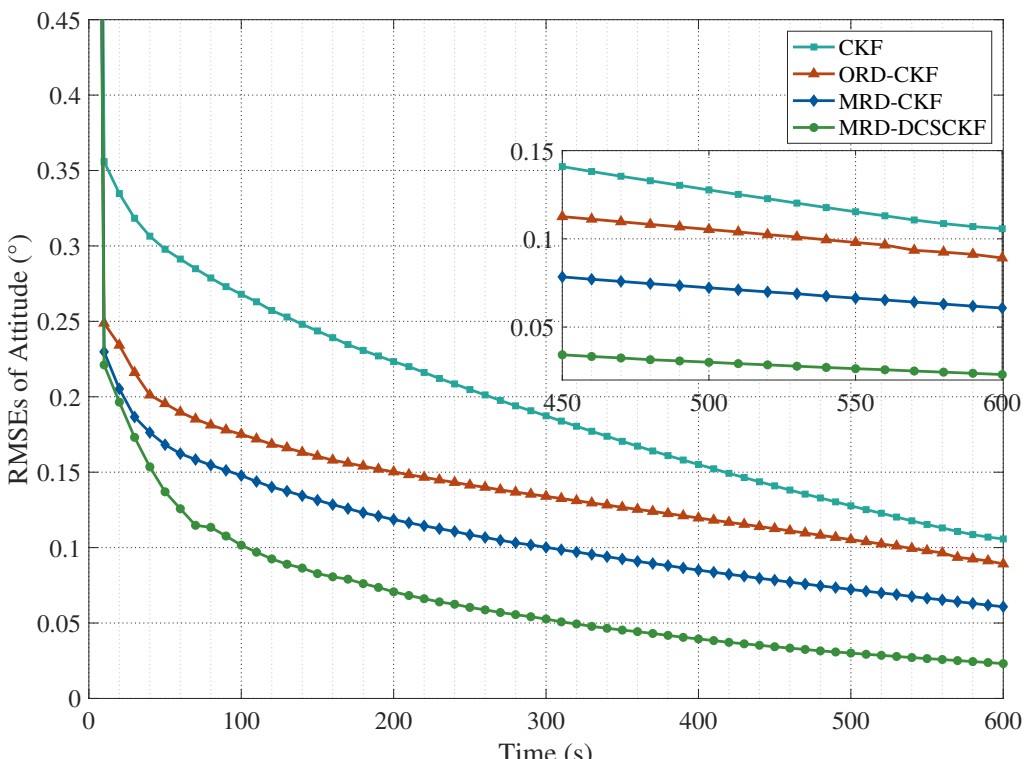

**Figure 10.** RMSEs of the attitude with CKF, ORD-CKF, MRD-CKF, and MRD-DCSCKF.

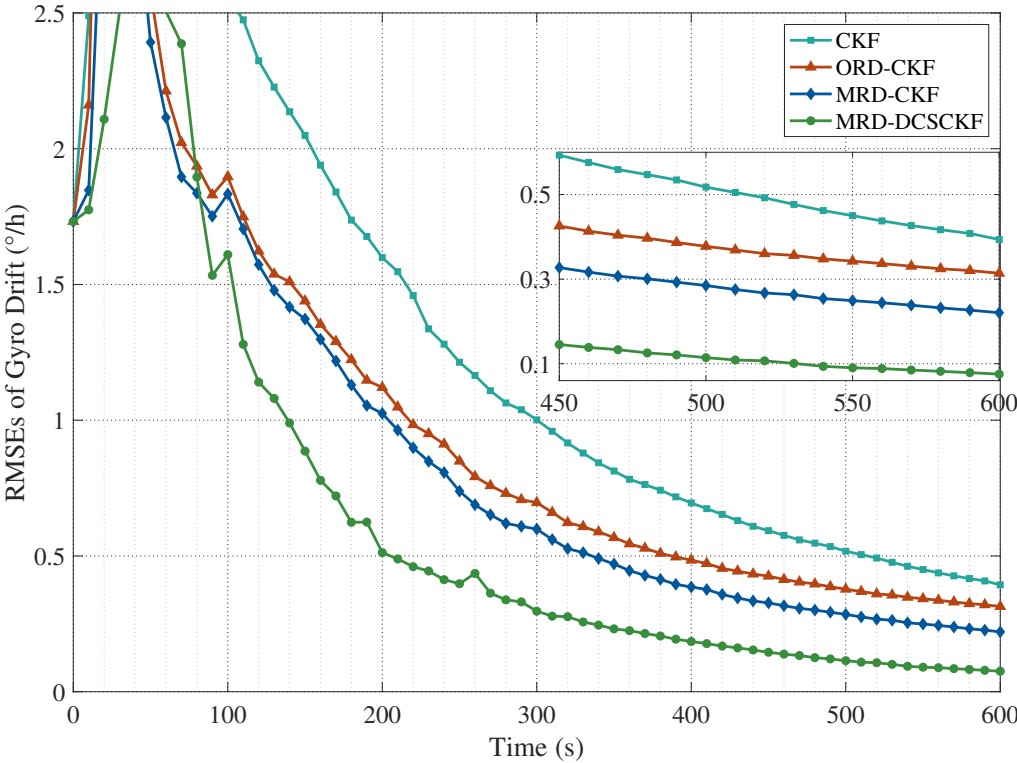

**Figure 11.** RMSEs of the gyro drift with CKF, ORD-CKF, MRD-CKF, and MRD-DCSCKF.

Figures 12–15 present the tracking errors of MRD-DCSCKF. As can be seen, the error curves converge quickly to the $3\sigma$ bounds, which are indicated by the dashed line. This demonstrates that the convergence property of MRD-DCSCKF is sufficient.

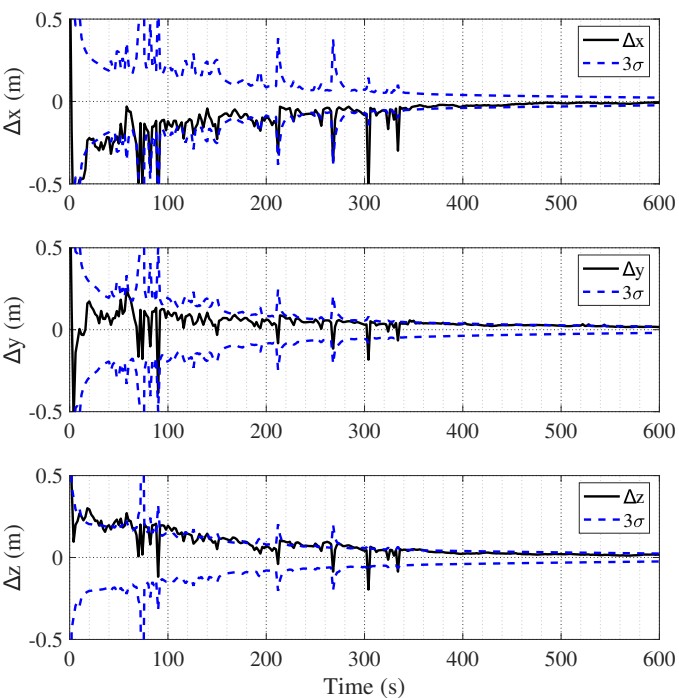

**Figure 12.** Position error.

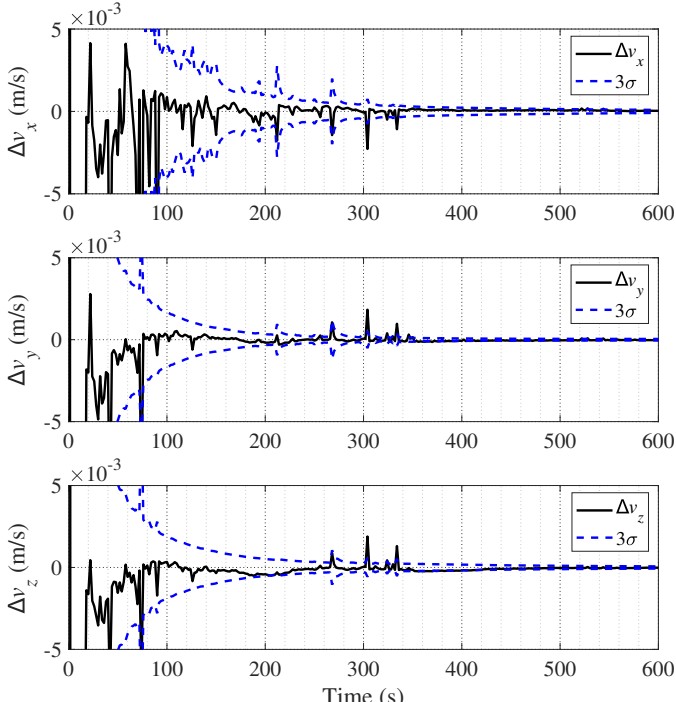

**Figure 13.** Velocity error.

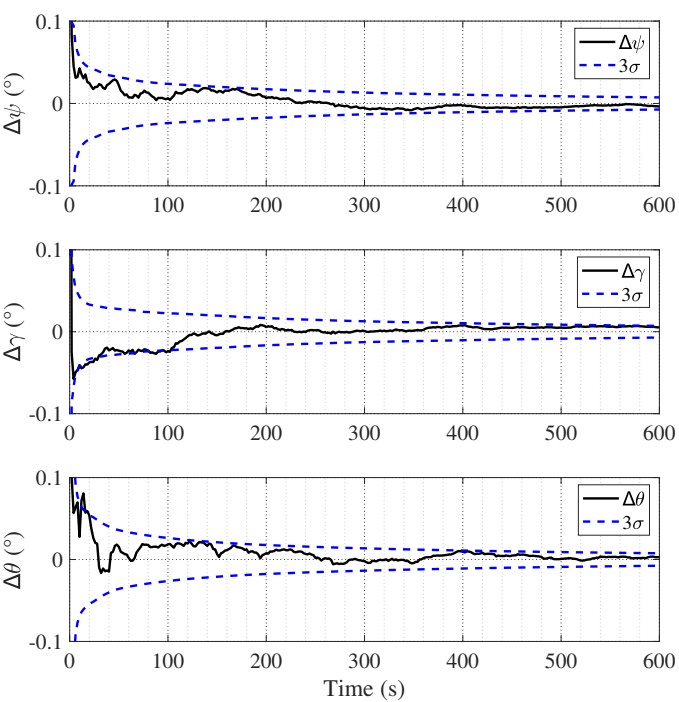

**Figure 14.** Attitude error.

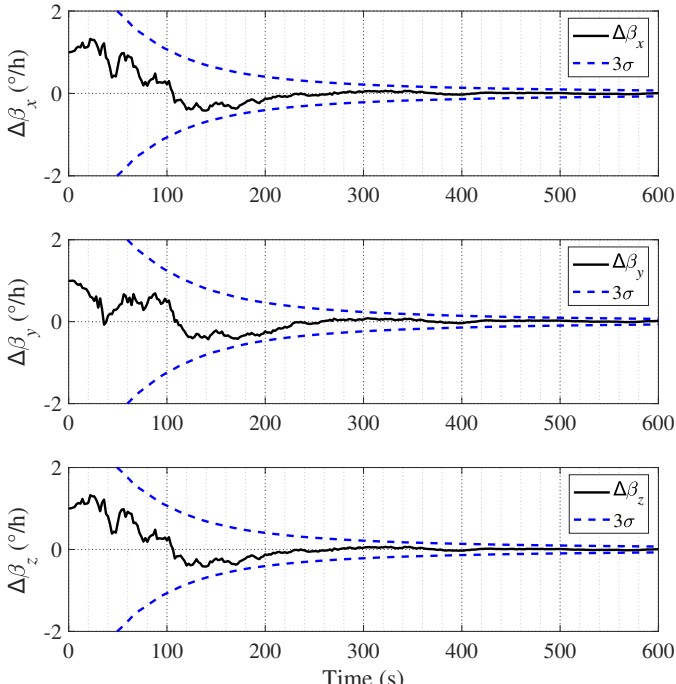

**Figure 15.** Gyro drift error.

Simulations to investigate the effects of delay probability on the MRD-DCSCKF were conducted, and Figures 16–19 show the ARMSEs from 450 to 600 s with delay probabilities of $0 \leq p \leq 0.2$. It is clear that MRD-DCSCKF achieved the highest estimation accuracy among the four filters.

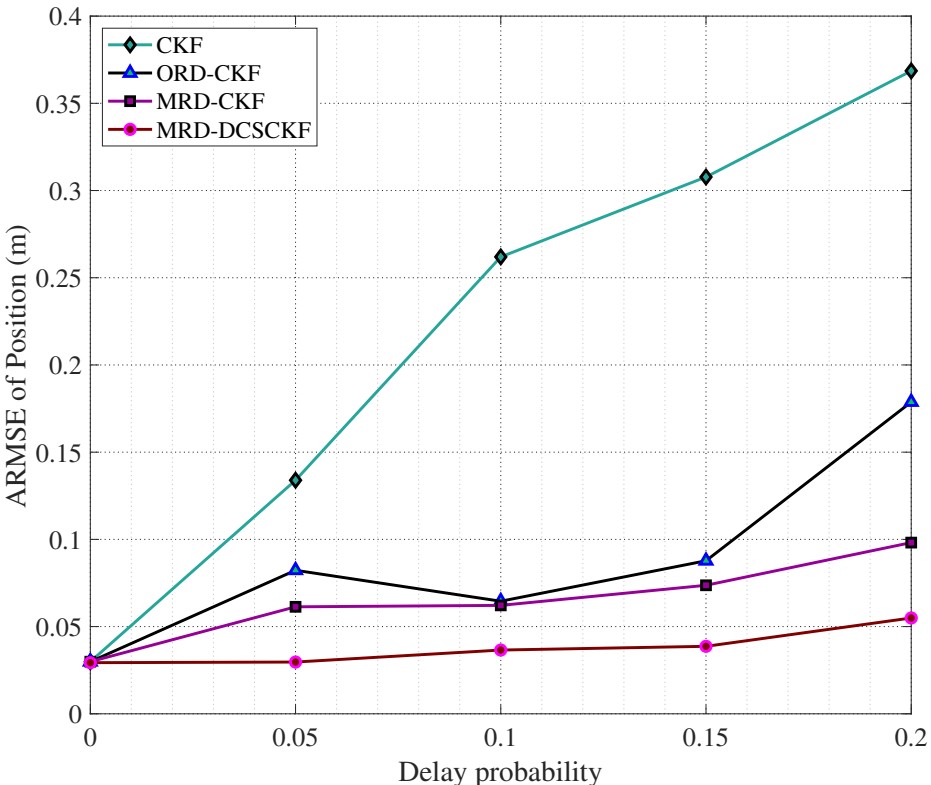

**Figure 16.** ARMSEs of position under different delay probabilities.

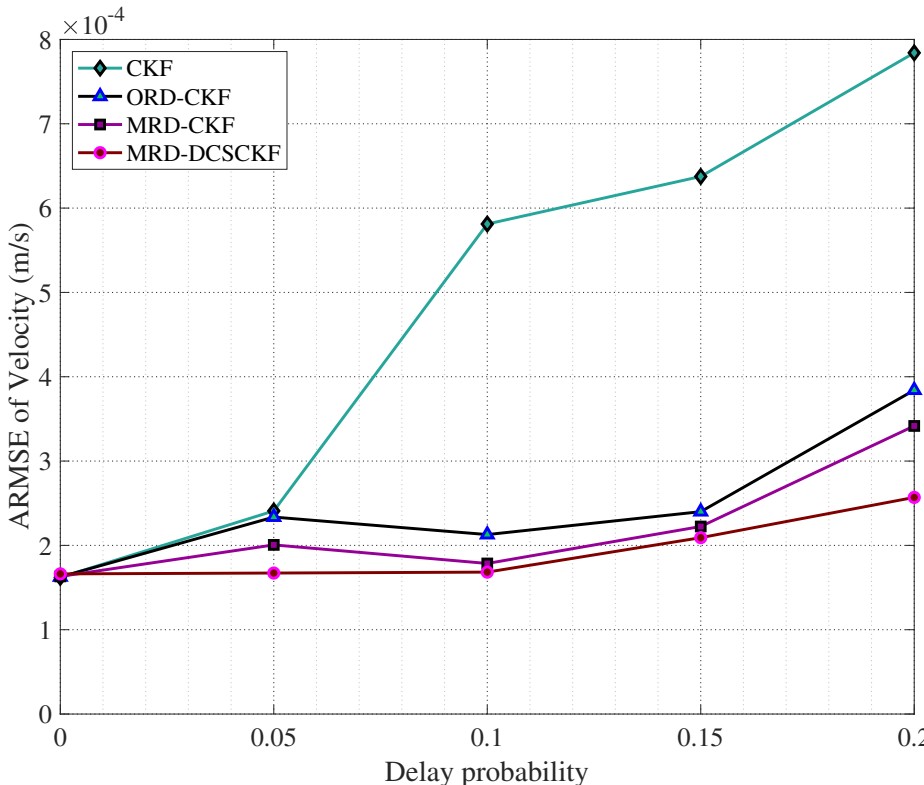

**Figure 17.** ARMSEs of velocity under different delay probabilities.

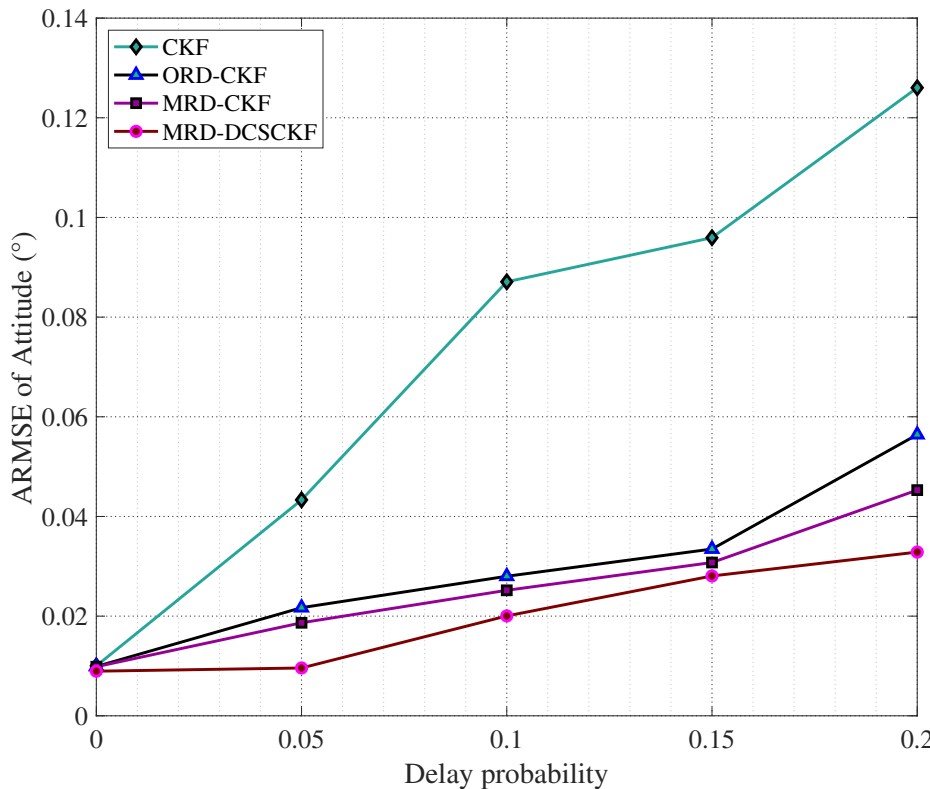

**Figure 18.** ARMSEs of attitude under different delay probabilities.

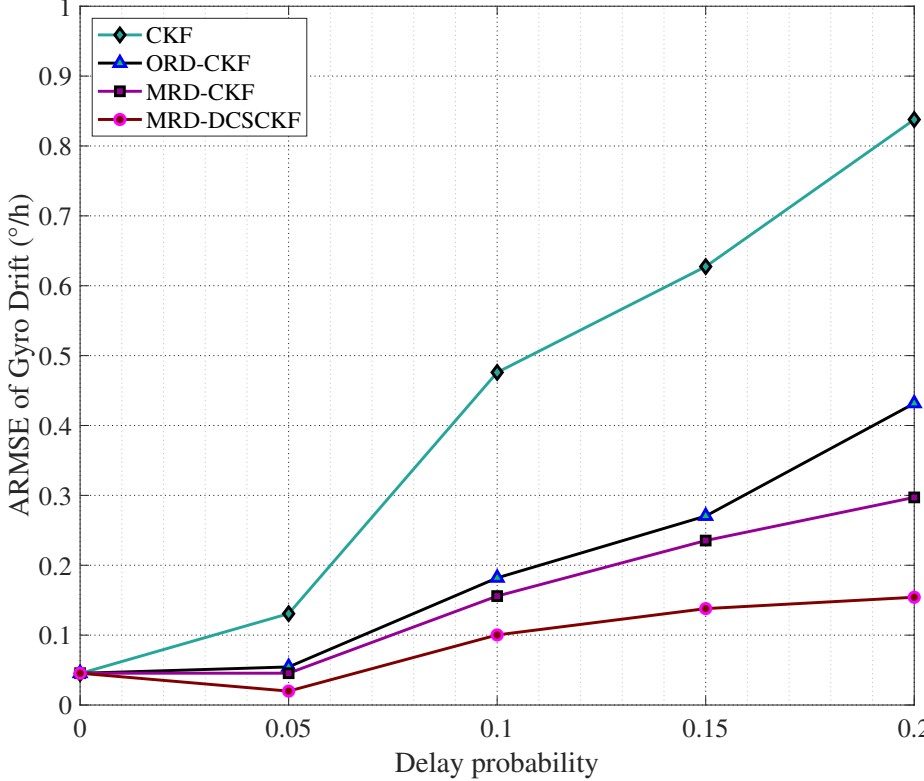

**Figure 19.** ARMSEs of gyro drift under different delay probabilities.

Figures 20–23 show the ARMSEs from 450 to 600 s with perturbing parameters of $0 \leq \varepsilon \leq 0.2$. As shown, the estimation accuracy of all the filters gradually decreases with increasing perturbing parameters. However, the MRD-DCSCKF exhibits superior robustness and performance of all the filters, even under these conditions.

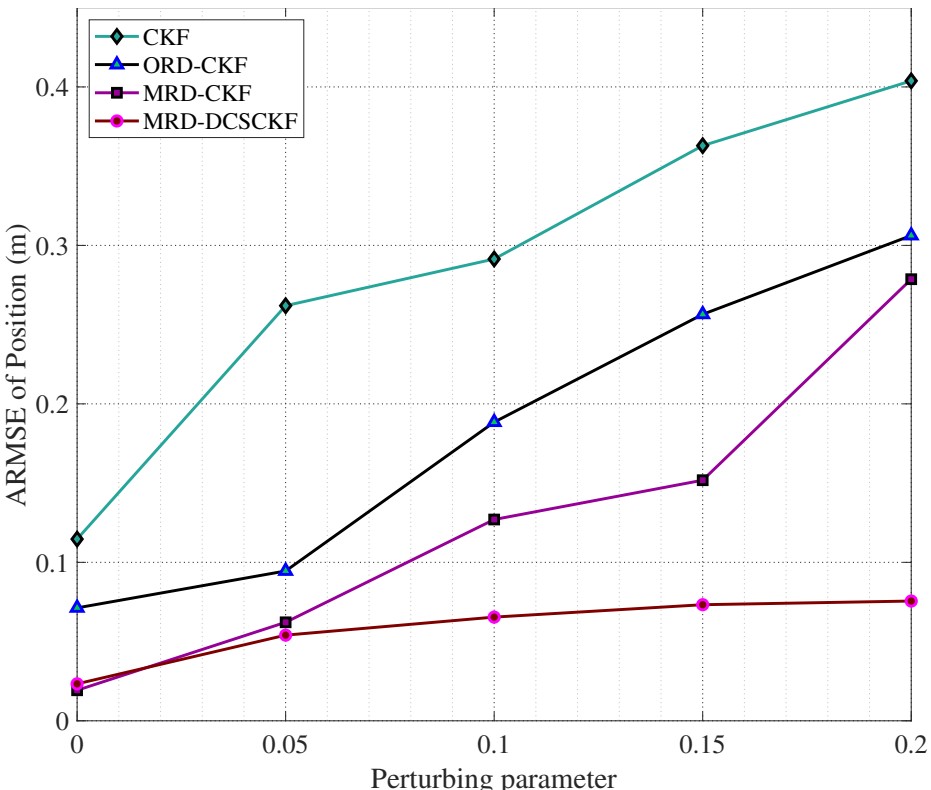

**Figure 20.** ARMSEs of position under different perturbing parameters.

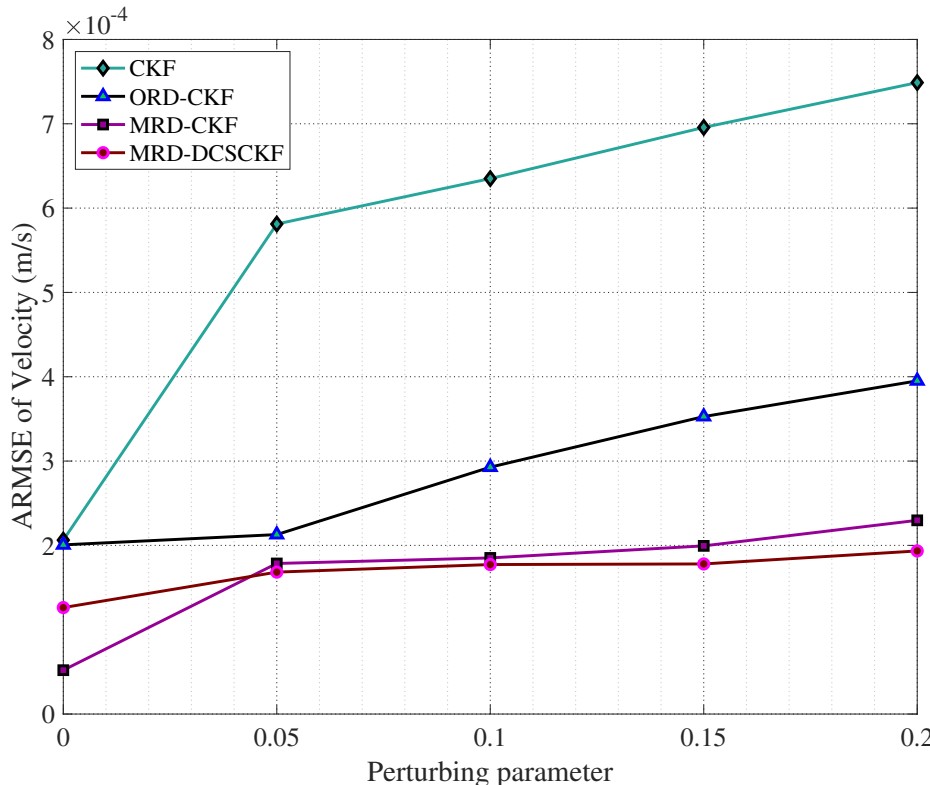

**Figure 21.** ARMSEs of velocity under different perturbing parameters.

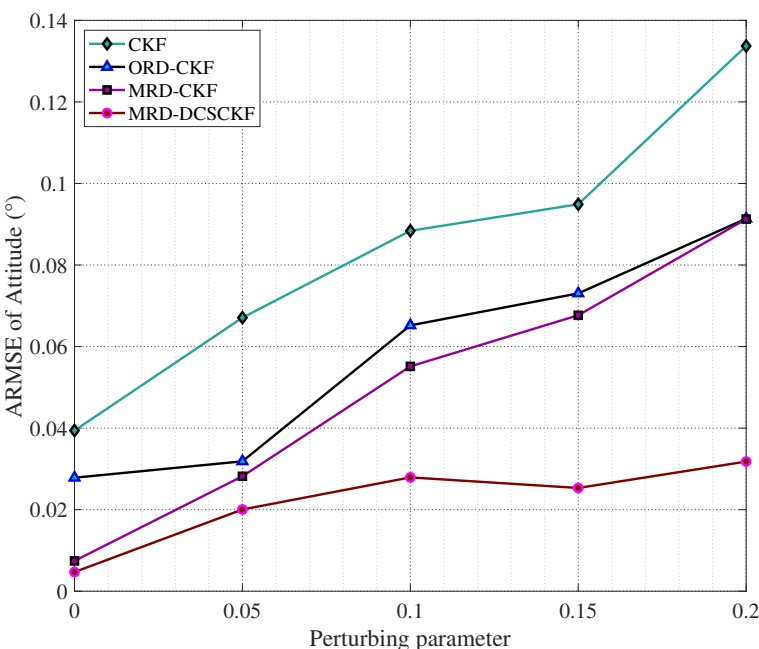

**Figure 22.** ARMSEs of attitude under different perturbing parameters.

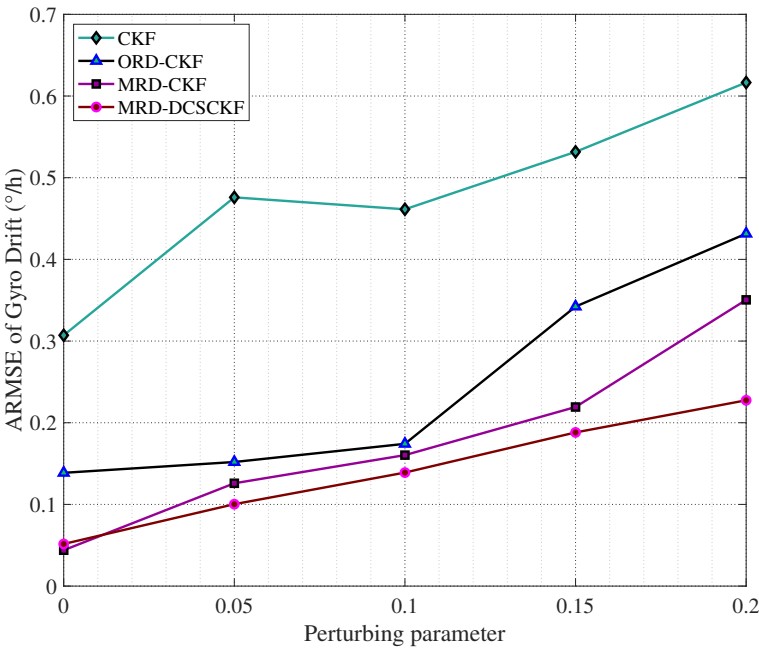

**Figure 23.** ARMSEs of gyro drift under different perturbing parameters.

## 7. Conclusions

This paper proposes a novel multiple-step randomly delayed, robust filter, referred to as the multiple-step, randomly delayed, dynamic-covariance-scaling cubature Kalman filter (MRD-DCSCKF), to effectively handle randomly delayed measurements and outliers. The MRD-DCSCKF uses a state-augmentation approach to break the limitations of the delayed steps and reformulates the state update equations of Kalman filter based on the delayed measurements modeled according to a set of Bernoulli random variables. Meanwhile, the proposed filter relies on dynamic-covariance-scaling, robust kernel to suppress measurement-outliers. The application of MRD-DCSCKF to vision-based spacecraft-relative navigation is investigated, where the relative dynamics are described with the T-H equations, and quaternions and generalized Rodrigues parameters are introduced to estimate spacecraft relative attitudes. The simulation results illustrate that the MRD-DCSCKF is able

to precisely estimate the status of the spacecraft with high precision, even with randomly delayed measurements and outliers, as compared to other algorithms.

**Author Contributions:** Conceptualization, Y.C.; methodology, Y.C.; software, Y.C.; validation, Y.C.; formal analysis, H.Z.; investigation, Y.C. and H.L.; data curation, Y.C., R.M. and H.Z.; writing—original draft preparation, Y.C.; writing—review and editing, Y.C., R.M., H.Z. and H.L.; supervision, R.M.; All authors have read and agreed to the published version of the manuscript.

**Funding:** This research received no external funding.

**Institutional Review Board Statement:** Not applicable.

**Informed Consent Statement:** Not applicable.

**Data Availability Statement:** Not applicable.

**Conflicts of Interest:** The authors declare no conflict of interest.

## Abbreviations

The following abbreviations are used in this manuscript:

| | |
|---|---|
| VISNAV | Vision-based navigation |
| DCS | Dynamic covariance scaling |
| BRV | Bernoulli random variable |
| PSD | Position sensing diode |
| LED | Light-emitting diode |
| SLAM | Simultaneous localization and mapping |
| ECI | Earth-centered inertial |
| LVLH | Local–vertical–local–horizontal |
| GRP | Generalized Rodrigues parameter |
| MCC | Maximum correntropy criterion |
| SE | State estimation |
| GA | Gaussian approximation |
| PDF | Probability density distribution |
| KF | Kalman filter |
| PF | Particle filter |
| EKF | Extended Kalman filter |
| UKF | Unscented Kalman filter |
| UT | Unscented transformation |
| CKF | Cubature Kalman filter |
| GGLQ | Generalized Gauss–Laguerre quadrature |
| HCKF | High-degree cubature Kalman filter |
| MAEKF | Modified adaptive extended Kalman filter |
| STF | Student's $t$ filter |
| ORD-CKF | One-step randomly delayed cubature Kalman filter |
| MRD-CKF | Multiple-step randomly delayed cubature Kalman filter |
| MRD-DCSCKF | Multiple-step randomly delayed dynamic-covariance-scaling cubature Kalman filter |

## Appendix A

*Appendix A.1*

According to the third-order the spherical-radial cubature rule [42], the evaluation of the CKF sampling points with $n$-dimensional state vector $x$ and covariance $P$ is computed, as follows.

$$\chi_i = \text{Trans}[x, P] = x + S\varsigma_i, i = 1, 2, \cdots, 2n \tag{A1}$$

where $S$ is the square root of $P\ (P = S \cdot S^T)$. $\varsigma_i$ is an element in the $2n$ cubature points, and its weight is $\omega_i = \frac{1}{2n}$. In addition, $\{\varsigma_i\}$ is as follows

$$\sqrt{n}\left[\begin{pmatrix} 1 \\ 0 \\ \vdots \\ 0 \end{pmatrix}, \cdots, \begin{pmatrix} 0 \\ 0 \\ \vdots \\ 1 \end{pmatrix}, \begin{pmatrix} -1 \\ 0 \\ \vdots \\ 0 \end{pmatrix}, \cdots, \begin{pmatrix} 0 \\ 0 \\ \vdots \\ -1 \end{pmatrix}\right] \tag{A2}$$

*Appendix A.2*

The sub-update of the multiple-step randomly delayed cubature Kalman filter [25] at the $s$-th step is calculated as below

$$K_k^s = P_{k/k-1}^{Xy,s}\left(P_{k/k-1}^{yy,s} + R_{k-s}\right)^{-1} \tag{A3}$$

$$\hat{X}_{k/k}^s = \hat{X}_{k/k-1} + K_k^s\left(y_k - \hat{y}_{k/k-1}^s\right), \ s = 0, 1, 2, \ldots, d \tag{A4}$$

$$P_{k/k}^s = P_{k/k-1} - K_k^s\left(P_{k/k-1}^{Xy,s}\right)^T, \ s = 0, 1, 2, \ldots, d \tag{A5}$$

*Appendix A.3*

The combination of CKF and the M-estimator results in a robust CKF, which adopts different robust kernel functions to suppress outliers [30,43]. The state updates of robust CKF are the same as CKF, and its measurement updates through constructing a linear regression model with the prior estimation of the filter and the measurement model. Finally, it completes the robust update process by iteration. This section briefly reviews the measurement updates of the linear regression robust CKF.

The measurement equation is approximated as follows:

$$y_k \approx h\left(\hat{x}_{k|k-1}\right) + H_k\left(x_k - \hat{x}_{k|k-1}\right) + w_k \tag{A6}$$

The state prediction error $\delta_k$ is denoted as following:

$$\delta_k = x_k - \hat{x}_{k|k-1} \tag{A7}$$

Then, the regression problem takes the form

$$\left[\begin{matrix} y_k - h\left(\hat{x}_{k|k-1}\right) + H_k\hat{x}_{k|k-1} \\ \hat{x}_{k|k-1} \end{matrix}\right] = \left[\begin{matrix} H_k \\ I \end{matrix}\right]x_k + \left[\begin{matrix} w_k \\ -\delta_k \end{matrix}\right] \tag{A8}$$

where $H_k = \left[\left(P_{k|k-1}\right)^{-1}(P^{xy})\right]^T$ is the measurement matrix.

Some quantities are given as

$$S_k = \left[\begin{matrix} R_k & 0 \\ 0 & P_{k|k-1} \end{matrix}\right] \tag{A9}$$

$$\widehat{z}_k = S_k^{-1/2}\left[\begin{matrix} y_k - h\left(\hat{x}_{k|k-1}\right) + H_k\hat{x}_{k|k-1} \\ \hat{x}_{k|k-1} \end{matrix}\right] \tag{A10}$$

$$M_k = S_k^{-1/2}\left[\begin{matrix} H_k \\ I \end{matrix}\right] \tag{A11}$$

$$\boldsymbol{\xi}_k = \boldsymbol{S}_k^{-1/2} \begin{bmatrix} \boldsymbol{w}_k \\ -\boldsymbol{\delta}_k \end{bmatrix} \tag{A12}$$

Then Equation (A8) is transformed to

$$\widehat{\boldsymbol{z}}_k = \boldsymbol{M}_k \boldsymbol{x}_k + \boldsymbol{\xi}_k \tag{A13}$$

where $\boldsymbol{\xi}_k$ is residual error with $\boldsymbol{\xi}_k = \boldsymbol{M}_k \boldsymbol{x}_k - \widehat{\boldsymbol{z}}_k$ .

Minimize the following cost function to solve above-mentioned regression problem:

$$J(\boldsymbol{x}_k) = \sum_{i=1}^{m+n} \rho(\boldsymbol{\xi}_k) \tag{A14}$$

where $\rho(\cdot)$ is the cost function. $m$ denotes the dimension of measurement $\boldsymbol{y}_k$.

The solution of Equation (A14) satisfies

$$\sum_{k=1}^{m+n} \varphi(\boldsymbol{\xi}_k) \frac{\partial \boldsymbol{\xi}_k}{\partial \boldsymbol{x}_k} = 0 \tag{A15}$$

where $\varphi(\boldsymbol{\xi}_k) = \rho'(\boldsymbol{\xi}_k)$ . $\psi(\cdot)$ is the weight function defined as $\psi(\boldsymbol{\xi}_k) = \frac{\varphi(\boldsymbol{\xi}_k)}{\boldsymbol{\xi}_k}$ , and the corresponding weight matrix is $\boldsymbol{\Psi} = diag[\psi(\boldsymbol{\xi}_i)]$ . $\boldsymbol{\xi}_{i,k}$ is the $i$-th component of $\boldsymbol{\xi}_k$ . Equation (A15) could be written in matrix form, as follows:

$$\boldsymbol{M}_k^T \boldsymbol{\Psi} (\boldsymbol{M}_k \boldsymbol{x}_k - \widehat{\boldsymbol{z}}_k) = 0 \tag{A16}$$

The solution of Equation (A16) can be obtained after $j$ times iteration, as follows:

$$\boldsymbol{x}_k^{(j+1)} = \left( \boldsymbol{M}_k^T \boldsymbol{\Psi}^{(j)} \boldsymbol{M}_k \right)^{-1} \boldsymbol{M}_k^T \boldsymbol{\Psi}^{(j)} \widehat{\boldsymbol{z}}_k \tag{A17}$$

The state estimation covariance could be computed by the following:

$$\boldsymbol{P}_{k|k} = \left( \boldsymbol{M}_k^T \boldsymbol{\Psi}^{(j)} \boldsymbol{M}_k \right)^{-1} \tag{A18}$$

In this paper, the calculation process of Equation (A8)-Equation (A16) is summarized as follows:

$$\left[ \boldsymbol{M}, \widehat{\boldsymbol{z}} \right] = \text{Robust\_update} \left[ \boldsymbol{y}_k, h(\cdot), \boldsymbol{P}_{k/k-1}, \boldsymbol{P}^{xy}, \hat{\boldsymbol{x}}_{k|k-1}, \boldsymbol{R} \right] \tag{A19}$$

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
