# Peer review of "A Multiple-Step, Randomly Delayed, Robust Cubature Kalman Filter for Spacecraft-Relative Navigation"

_aerospace, doi:10.3390/aerospace10030289_

Round 1
Reviewer 1 Report
To calculate the unbiased root mean square error (RMSE) in formula (67),
it is necessary to replace the factor M^{-1} with (M-1)^{-1}
Reviewer 2 Report
The paper presents a navigation architecture for delayed measurements and contaminated Gaussian distribution with application to spacecraft relative navigation.
- The paper presents a novel filtering technique that is more robust compared to a Kalman filter and other traditional techniques.
- The paper also shows a good application of the filter to a spacecraft navigation problem.
- There are some minor typos in the paper. Please go over them.
- The introduction does not flow really well. For example, The author talks about the Kalman filter, then jump to VISNAV, then talk about different sensors, etc. This can be fixed by talking about the problem, then going over the current solutions, then why the existing solutions do not work, and then talking about your proposed solution. In short, the introduction needs to be fixed.
- The grids on the plots don't appear correctly. Please fix this.
- Figure 5 is too large.
- Overall, I think paper shows a novel development of a filter. However, a lot of work is built from ref. 29.
- Paper should be ready to be published after addressing these minor comments and grammatical errors.
Reviewer 3 Report
The paper proposes a solution to the problem of delayed measurements and contaminated Gaussian distribution in navigation systems. The proposed filter, called multiple-step randomly delayed dynamic-covariance-scaling cubature Kalman filter (MRD-DCSCKF), has been applied to spacecraft relative navigation simulations and has shown superior performance compared to other algorithms, providing high-accuracy position and attitude estimation. Overall, the manuscript is well written and according to this reviewer the work is worth for publication. Here, some minor comments:
o Line 3: we proposes ïƒ we propose
o Line 14: Can you provide slightly more context about the Monte Carlo simulations? In particular, it would be appropriate to specify which model parameter is varied in the Monte Carlo analysis.
o Line 20: here, a lot of examples are made regarding the use of high precision position and attitude estimation, but only one reference is cited; I would suggest to add some more references covering the different cases (just a few examples spanning over various cases: https://doi.org/10.1109/MetroAeroSpace51421.2021.9511774 , https://doi.org/10.1016/j.actaastro.2022.11.041 ,
https://doi.org/10.1002/rob.22138 )
o Line 26: In the sentence “It is usually adopted to determine the relative position and attitude of spacecraft within several hundred meters” I am not sure if “several hundreds of meters” is referred to the positioning accuracy or if it is the relative distance between the two spacecraft. In the first case, it would be appropriate to mention also the accuracy for attitude determination. Please, provide also some references which point out the attainable accuracy.
o Line 29: The sentence “Since PSD possesses the merits of measuring the intensity and position of a light-point simultaneously” is a bit confusing, I am not sure that “since” is appropriate at the beginning of the phrase. I would suggest replacing it with: “PSD possesses the merits of measuring the intensity and position of a light-point simultaneously”.
o Line 38: The LVLH system is referred to the chief spacecraft or to the deputy spacecraft?
o Line 53: I would suggest rephrasing: “… can cause the measurements of sensors to be complicated” ïƒ “… degrade the measurements’ quality”
o Line 62: “Vision-based spacecraft relative navigation uses the optical camera to obtain measurements, which indicates that the measurement equations are nonlinear.” The subject of the sentence is not clear. An alternative could be “Vision-based spacecraft relative navigation uses the optical camera to obtain measurements, which are modeled with non-linear equations.”
o Line 74: Please, define the Genz’s code, or provide some references.
o Line 95: Differently from the other acronyms, it seems to me that BRV is not defined in the text but only in the acronym section.
o Line 101: research ïƒ researches
o Line 103: the relevant –> relevant
o Line 106: informations ïƒ information
o Line 142: As shown in Figure 1, it presents ïƒ Figure 1 represents …
o Line 145: Due to the data transmission channel is not reliable, there is one-step or multiple-step delay in the measurements received from sensor 1, where the solid line indicates that the measurements are synchronized and the dashed line indicates that the measurements are delayed.” Due to the unreliability of the data transmission channel …
o Figure 2: Please, adjust the label adding details. Example: “Gaussian distribution (red) and contaminated Gaussian distribution (black) with ε = 0.05 , σ1 = 1 and σ2 = 7.5σ1”
o Line 193:
has not been previously defined
o Line 207: Calculate ïƒ the measurements …. can be calculated as
o Line 218: is as follows ïƒ are as follows
o Line 224: Compute ïƒ the … can be computed as:
o Line 226: Equation 43 shall be introduced.
o Line 252: the chief and deputy ïƒ the chief and deputy spacecraft
o Line 275: to represents ïƒ to represent
o Line 283: Can you quantify what is meant by “close”?
o Line 286: The acronym VISNAV has already been defined
o Section 6.1: Wouldn’t be worth to represent/give indications also on the orbit of the deputy spacecraft?
o Line 313: It is not clear to me what is varying among the different simulations of the Monte Carlo analysis?
o Line 324: “since” does not seem adequate in this sentence
o Figure 8: at 100 s, and between 100 and 200 s, seems that there are irregular variations of the RMSE with respect to the other methods; can you motivate this behavior?
o Figure 9: do you know why the MRD-DCSKF become better than the other filter only after 350s?
o Line 379: review ïƒ reviews
o Line 383: “then” is repeated two times
Reviewer 4 Report
The paper “A multiple-step randomly delayed robust cubature Kalman filter for spacecraft relative navigation” is devoted to relative state estimation algorithm development and application to formation flying. The topic of the paper can be interesting for specialists in relative motion determination and control. The paper is well-written, though some statements in the text are not clear and some additional comments are required. There are several suggestions.
It is not clear what the authors mean by “ideal” measurements in line 132? The problem statement should be described in more details. Is it correct that the actual measurements (3) belong to the “state estimation point”, though the time obtained measurement is not known and random (according to Figure 1)? What can be the nature of the not known delay? For optical sensors the time of sampling is recorded and the measurement is processed using this information of the actual time of the data obtained.
The cases of “non-occurrence delay” and of “occurrence delay” are drawn separately in Fig. 4, though in the text there is no details on some special difference in algorithm expressions between these cases. The authors should emphasize the features of the two cases.
In the state vector includes the generalized Rodrigues parameters and the angular velocity bias along with the relative transitional position and velocity vectors. The problem statement is not clear in this case. To estimate the relative attitude using the kinematics equations (56) the angular velocity sensor measurements from both the deputy and the chief satellite are required. How can the relative attitude be estimated using only the deputy angular velocity sensor?
The Fig. 6 is not clear. Why the beacon is placed separately from the cube? What does the cube represent? If it is the chief satellite, why is it not located near the chief reference frame?
